# NONPARAMETRIC NEURAL NETWORKS

**George Philipp, Jaime G. Carbonell**
Carnegie Mellon University
Pittsburgh, PA 15213, USA
`george.philipp@email.de; jgc@cs.cmu.edu`

## ABSTRACT

Automatically determining the optimal size of a neural network for a given task without prior information currently requires an expensive global search and training many networks from scratch. In this paper, we address the problem of automatically finding a good network size during a single training cycle. We introduce *nonparametric neural networks*, a non-probabilistic framework for conducting optimization over all possible network sizes and prove its soundness when network growth is limited via an $\ell_p$ penalty. We train networks under this framework by continuously adding new units while eliminating redundant units via an $\ell_2$ penalty. We employ a novel optimization algorithm, which we term "Adaptive Radial-Angular Gradient Descent" or *AdaRad*, and obtain promising results.

## 1 INTRODUCTION

Automatically choosing a neural network model for a given task without prior information is a challenging problem. Formally, let $\Theta$ be the space of all models considered. The goal of *model selection* is then, usually, to find the value of the *hyperparameter* $\theta \in \Theta$ that minimizes a certain criterion $c(\theta)$, such as the validation error achieved by the model represented by $\theta$ when trained to convergence. Because $\Theta$ is large, structured and heterogeneous, $c$ is complex, and gradients of $c$ are generally not available, the most popular methods for optimizing $c$ perform zero-order, black-box optimization and do not use any information about $c$ except its value for certain values of $\theta$. These methods select one or more values of $\theta$, compute $c$ at those values and, based on the results, select new values of $\theta$ until convergence is achieved or a time limit is reached. The most popular such methods are grid search, random search (e.g. Bergstra & Bengio (2012)) and Bayesian optimization using Gaussian processes (e.g. Snoek et al. (2012)). Others utilize random forests (Hutter et al., 2009), deep neural networks (Snoek et al., 2015) and recently Bayesian neural networks (Springenberg et al., 2016) and reinforcement learning (Zoph & Le, 2017).

These black-box methods have two drawbacks. (A) To obtain each value of $c$, they execute a full network training run. Each run can take days on many cores or multiple GPUs. (B) They do not exploit opportunities to improve the value of $c$ further by altering $\theta$ during each training run. In this paper, we present a framework we term *nonparametric neural networks* for selecting network size. We dynamically and automatically shrink and expand the network as needed to select a good network size during a single training run. Further, by altering network size during training, the network ultimately chosen can achieve a higher accuracy than networks of the same size that are trained from scratch and, in some cases, achieve a higher accuracy than is possible by black-box methods.

There has been a recent surge of interest in eliminating unnecessary units from neural networks, either during training or after training is complete. This strategy is called *pruning*. Alvarez & Salzmann (2016) utilize an $\ell_2$ penalty to eliminate units and Molchanov et al. (2017) compare a variety of strategies, whereas Figurnov et al. (2016) focuses on thinning convolutional layers in the spatial dimensions. While some of these methods even allow some previously pruned units to be added back in (e.g. Feng & Darrell (2015)), all of these strategies require a high-performing network model as a starting point from which to prune, something that is generally only available in well-studied vision and NLP tasks. We do not require such a starting point in this paper.

In section 2, we introduce the nonparametric framework and state its theoretical soundness, which we prove in section 7.1. In section 3, we develop the machinery for training nonparametric networks,

including a novel normalization layer in section 3.2, CapNorm, and a novel training algorithm in section 3.3, AdaRad. We provide experimental evaluation and analysis in section 4, further relevant literature in section 5 and conclude in section 6.

## 2 NONPARAMETRIC NEURAL NETWORKS

For the purpose of this section, we define a parametric neural network as a function $f(x) = \sigma_L.(\sigma_{L-1}.(..\sigma_2.(\sigma_1.(xW_1)W_2)..)W_L)$ of a $d_0$-dimensional row vector $x$, where $W_l \in \mathbb{R}^{d_{l-1}*d_l}, 1 \leq l \leq L$ are dense weight matrices of fixed dimension and $\sigma_l : \mathbb{R} \to \mathbb{R}, 1 \leq l \leq L$ are fixed non-linear transformations that are applied elementwise, as signified by the .() operator. The number of layers $L$ is also fixed. Further, the weight matrices are trained by solving the mini-mization problem $\min_{\mathbf{W}=(W)_l} \frac{1}{|D|} \sum_{(x,y)\in D} e(f(\mathbf{W}, x), y) + \Omega(\mathbf{W})$, where $D$ is the dataset, $e$ is an error function that consumes a vector of fixed size $d_L$ and the label $y$, and $\Omega$ is the regularizer.

We define a nonparametric neural network in the same way, except that the dimensionality of the weight matrices is undetermined. Hence, the optimization problem becomes

$$\min_{\mathbf{d}=(d)_l, d_l\in\mathbb{Z}_+, 1\leq l\leq L-1} \min_{\mathbf{W}=(W)_l, W_l\in\mathbb{R}^{d_{l-1}*d_l}, 1\leq l\leq L} \frac{1}{|D|} \sum_{(x,y)\in D} e(f(\mathbf{W}, x), y) + \Omega(\mathbf{W}) \quad (1)$$

Note that the dimensions $d_0$ and $d_L$ are fixed because the data and the error function $e$ are fixed. The parameter value now takes the form of a pair $(\mathbf{d}, \mathbf{W})$.

There is no guarantee that optimization problem 1 has a global minimum. We may be able to reduce the value of the objective further and further by using larger and larger networks. This would be problematic, because as networks become better and better with regards to the objective, they would become more and more undesirable in practice. It turns out that in an important case, this degeneration does not occur. Define the *fan-in regularizer* $\Omega_{in}$ and the *fan-out regularizer* $\Omega_{out}$ as

$$\Omega_{in}(\mathbf{W}, \lambda, p) = \lambda \sum_{l=1}^{L} \sum_{j=1}^{d_l} ||[W_l(1,j), W_l(2,j), .., W_l(d_{l-1}, j)]||_p \quad (2)$$

$$\Omega_{out}(\mathbf{W}, \lambda, p) = \lambda \sum_{l=1}^{L} \sum_{i=1}^{d_{l-1}} ||[W_l(i,1), W_l(i,2), .., W_l(i, d_l)]||_p \quad (3)$$

In plain language, we either penalize the incoming weights (*fan-in*) of each unit with a $p$-norm, or the outgoing weights (*fan-out*) of each unit. We now state the core theorem that justifies our formulation of nonparametric networks. The proof is found in the appendix in section 7.1.

**Theorem 1.** *Nonparametric neural networks achieve a global training error minimum at some finite dimensionality when $\Omega$ is a fan-in or a fan-out regularizer with $\lambda > 0$ and $1 \leq p < \infty$.*

## 3 TRAINING NONPARAMETRIC NETWORKS

Training nonparametric networks is more difficult than training parametric networks, because the space over which we optimize the parameter $(\mathbf{d}, \mathbf{W})$ is no longer a space of form $\mathbb{R}^d$, but is an infinite, discrete union of such spaces. However, we would still like to utilize local, gradient-based search. We notice, like (Wei et al., 2016), that there are pairs of parameter values with different dimensionality that are still in some sense "close" to one another. Specifically, we say that two parameter values $(\mathbf{d}_1, \mathbf{W}_1)$ and $(\mathbf{d}_2, \mathbf{W}_2)$ are *f-equivalent* if $\forall x \in \mathbb{R}^{d_0}, f(\mathbf{W}_1, x) = f(\mathbf{W}_2, x)$ where not necessarily $\mathbf{d}_1 = \mathbf{d}_2$. During iterative optimization, we can "jump" between those two parameter values while maintaining the output of $f$ and thus preserving locality. We define a *zero unit* as any unit for which either the fan-in or fan-out or both are the zero vector. Given any parameter value, the most obvious way of generating another parameter value that is $f$-equivalent to it is to add a zero unit to any hidden layer $l$ where $\sigma_l(0) = 0$ holds. Further, if we have a parameter value

that already contains a zero unit in such a hidden layer, removing it yields an $f$-equivalent parameter value.

Thus, we will use the following strategy for training nonparametric networks. We use gradient-based methods to adjust $\mathbf{W}$ while periodically adding and removing zero units. We use only nonlinearities that satisfy $\sigma(0) = 0$. It should be noted that while adding and removing zero units leaves the output of $f$ invariant, it does change the value of the fan-in and fan-out regularizers and thus the value of the objective. While it is possible to design regularizers that do not penalize such zero units, this is highly undesirable as it would stifle the regularizers ability to "reign in" the growth of the network during training.

To be able to reduce the network size during training, we must produce zero units and, it turns out, the fan-in and fan-out regularizers naturally produce such units as they induce sparsity, i.e. they cause individual weights to become exactly zero. This is well studied under the umbrella of sparse regression (see e.g. Tibshirani (1996)). The cases $p = 1$ and $p = 2$ are especially attractive because it is computationally convenient to integrate them into a gradient-based optimization framework via a shrinkage / group shrinkage operator respectively (see e.g. Back & Teboulle (2006)). Further, $p = 1$ and $p = 2$ differ in their effect on the parameter value. $p = 1$ sets individual weights to zero and thus leads to sparse fan-ins and fan-outs and thus ultimately to sparse weight matrices. A unit can only become a zero unit if each weight in its fan-in or each weight in its fan-out has been set to zero individually. $p = 2$, on the other hand, sets entire fan-ins (for the fan-in regularizer) or fan-outs (for the fan-out regularizer) to zero at once. Once the resulting zero units are removed, we obtain dense weight matrices. (For a basic comparison of 1-norm and 2-norm regularizers, see Yuan & Lin (2006) and for a comparison in the context of neural networks, see Collins & Kohli (2014).) While there is recent interest in learning very sparse weight matrices (e.g. Guo et al. (2016)), current hardware is geared towards dense weight matrices (Wen et al., 2016). Hence, for the remainder of this paper, we will focus on the case $p = 2$. Further, we will focus on the fan-in rather than the fan-out regularizer.

When a new zero unit is added, we must choose its fan-in and fan-out. While one of the two weight vectors must be zero, the other can have an arbitrary value. We make the simple choice of initializing the other weight vector randomly. Since we are going to use the fan-in regularizer, we will initialize the fan-out to zero and the fan-in randomly. This will give each new unit the chance to learn and become useful before the regularizer can shrink its fan-in to zero. If it does become zero nonetheless, the unit is eliminated.

## 3.1 Self-similar Nonlinearities

For layers 1 through $L - 1$, it is best to use nonlinearities that satisfy $\sigma(cs) = c\sigma(s)$ for all $c \in \mathbb{R}_{\geq 0}$ and $s \in \mathbb{R}$. We call such nonlinearities *self-similar*. ReLU (Dahl et al., 2013) is an example of this. Self-similarity also implies $\sigma(0) = 0$.

Recall that the fan-in and fan-out regularizers shrink the values of weights during training. This in turn affects the scale of the values to which the nonlinearities are applied. (These values are called *pre-activations*.) The advantage of self-similar nonlinearities is that this change of scale does not affect the shape of the feature.

In contrast, the impact of a nonlinearity such as $\tanh$ on pre-activations varies greatly based on their scale. If the pre-activations have very large absolute values, $\tanh$ effectively has a binary output. If they have very small absolute values, $\tanh$ mimics a linear function. In fact, all nonlinearities that are differentiable at 0 behave approximately like a linear function if the pre-activations have sufficiently small absolute values. This would render the unit ineffective. Since we expect some units to have small pre-activations due to shrinkage, this is undesirable.

By being invariant to the scale of pre-activations, self-similar nonlinearities further eliminate the need to tune how much regularization to assign to each layer. This is expressed in the following proposition which is proved in section 7.2.

**Proposition 1.** *If all nonlinearities in a nonparametric network model except possibly $\sigma_L$ are self-similar, then the objective function 1 using a fan-in or fan-out regularizer with different regularization parameters $\lambda_1, .., \lambda_L$ for each layer is equivalent to the same objective function using the single regularization parameter $\lambda = (\prod_{l=1}^{L} \lambda_l)^{\frac{1}{L}}$ for each layer, up to rescaling of weights.*

**1** **input**: $\alpha_r$: radial step size; $\alpha_\phi$: angular step size; $\lambda$: regularization hyperparameter; $\beta$: mixing
 rate; $\epsilon$: numerical stabilizer; $\mathbf{d}^0$: initial dimensions; $\mathbf{W}^0$: initial weights; $\nu$: unit addition
 rate; $\nu_{\text{freq}}$: unit addition frequency; $T$: number of iterations

**2** $\phi_{\max} = 0$; $c_{\max} = 0$; $\mathbf{d} = \mathbf{d}^0$; $\mathbf{W} = \mathbf{W}^0$;

**3** **for** $l = 1$ **to** $L$ **do**

**4** set $\bar{\phi}_l$ (angular quadratic running average) and $c_l$ (angular quadratic running average capacity)
 to zero vectors of size $d_l^0$;

**5** **end**

**6** **for** $t = 1$ **to** $T$ **do**

**7** set $D^t$ to mini-batch used at iteration $t$;

**8** $\mathbf{G} = \frac{1}{|D|} \nabla_\mathbf{W} \sum_{(x,y) \in D^t} e(f(\mathbf{W}, x), y)$;

**9** **for** $l = L$ **to** $1$ **do**

**10** **for** $j = d_l$ **to** $1$ **do**

**11** decompose $[G_l(i,j)]_i$ into a component parallel to $[W_l(i,j)]_i$ (call it $r$) and a
 component orthogonal to $[W_l(i,j)]_i$ (call it $\phi$) such that $[G_l(i,j)]_i = r + \phi$;

**12** $\bar{\phi}_l(j) = (1-\beta)\bar{\phi}_l(j) + \beta||\phi||_2^2$; $c_l(j) = (1-\beta)c_l(j) + \beta$;

**13** $\phi_{\max} = \max(\phi_{\max}, \bar{\phi}_l(j))$; $c_{\max} = \max(c_{\max}, c_l(j))$ ;

**14** $\phi_{\text{adj}} = \frac{\sqrt{\frac{\phi_{\max}}{c_{\max}}}}{\sqrt{\frac{\bar{\phi}_l(j)}{c_l(j)} + \epsilon}} \phi$;

**15** $[W_l(i,j)]_i = [W_l(i,j)]_i - \alpha_r r$ ;

**16** rotate $[W_l(i,j)]_i$ by angle $\alpha_\phi ||\phi_{\text{adj}}||_2$ in direction $-\frac{\phi_{\text{adj}}}{||\phi_{\text{adj}}||_2}$;

**17** shrink($[W_l(i,j)]_i, \alpha_r \lambda \frac{|D^t|}{|D|}$);

**18** **if** $l < L$ *and* $[W_l(i,j)]_i$ *is a zero vector* **then**

**19** remove column $j$ from $W_l$; remove row $j$ from $W_{l+1}$; remove element $j$ from $\bar{\phi}_l$
 and $c_l$; decrement $d_l$;

**20** **end**

**21** **end**

**22** **if** $t = 0 \mod \nu_{\text{freq}}$ **then**

**23** $\nu' = \nu$; `// if `$\nu \notin \mathbb{Z}$`, we can set e.g. `$\nu' = $`Poisson(`$\nu$`)`

**24** add $\nu'$ randomly initialized columns to $W_l$; add $\nu'$ zero rows to $W_{l+1}$; add $\nu'$ zero
 elements to $\bar{\phi}_l$ and $c_l$; $d_l = d_l + \nu'$;

**25** **end**

**26** **end**

**27** **end**

**28** **return** $\mathbf{W}$;

**Algorithm 1:** AdaRad with $\ell_2$ fan-in regularizer and the unit addition / removal scheme used in this paper in its most instructive (bot not fastest) order of computation. Note that $[]_i$ notation is used to indicate a vector over index $i$.

## 3.2 Capped batch normalization (*CapNorm*)

Recently, Ioffe & Szegedy (2015) proposed a strategy called batch normalization that quickly became the standard for keeping feed-forward networks well-conditioned during training. In our experiments, nonparametric networks trained without batch normalization could not compete with parametric networks trained with it. Batch normalization cannot be applied directly to nonparametric networks with a fan-in or fan-out regularizer, as it would allow us to shrink the absolute value of individual weights arbitrarily while compensating with the batch normalization layer, thus negating the regularizer. Hence, we make a small adjustment which results in a strategy we term *capped batch normalization* or *CapNorm*. We subtract the mean of the pre-activations of each hidden unit, but only scale their standard deviation if that standard deviation is greater than one. If it is less than one, we do not scale it. Also, after the normalization, we do not add or multiply the result with a free parameter. Hence, CapNorm replaces each pre-activation $z$ with $\frac{z-\mu}{\max(\sigma,1)}$, where $\mu$ is the mean and $\sigma$ is the standard deviation of that unit's pre-activations across the current mini-batch.

Table 1: Computational cost of efficient implementations of various algorithms, per mini-batch and weight. Operations that do not scale with the number of weights are not included. Operations associated with the computation of the gradient of the loss term (e.g. lines 7 and 8 in algorithm 1) as well as unit addition and removal (e.g. lines 18 to 24 in algorithm 1) are not included as they do not vary between algorithms.

| Algorithm | Network types | Cost per mini-batch and weight |
|---|---|---|
| SGD, no $\ell_2$ shrinkage | param., nonparam. | 1 multiplication |
| SGD with $\ell_2$ shrinkage | param., nonparam. | 3 multiplications |
| AdaRad, no $\ell_2$ shrinkage | param., nonparam. | 4 multiplications |
| AdaRad with $\ell_2$ shrinkage | param., nonparam. | 4 multiplications |
| RMSprop, no $\ell_2$ shrinkage | param. | 4 multiplications, 1 division, 1 square root |
| RMSprop with $\ell_2$ shrinkage | param. | 6 multiplications, 1 division, 1 square root |

### 3.3 ADAPTIVE RADIAL-ANGULAR GRADIENT DESCENT (*AdaRad*)

The staple method for training neural networks is stochastic gradient descent. Further, there are several popular variants: momentum and Nesterov momentum (Sutskever et al., 2013), AdaGrad (Duchi et al., 2011) and AdaDelta (Zeiler, 2012), RMSprop (Tieleman & Hinton, 2012) and Adam (Kingma & Ba, 2015). All of these methods center around two key principles: (1) averaging the gradient obtained over consecutive iterations to smooth out oscillations and (2) normalizing each component of the gradient so that each weight learns at roughly the same speed. Principle (2) turns out to be especially important for nonparametric neural networks. When a new unit is added, it does not initially contribute to the quality of the output of the network and so does not receive much gradient from the loss term. If the gradient is not normalized, that unit may take a very long time to learn anything useful. However, if we use a fan-in regularizer, we cannot normalize the components of the gradient outright as in e.g. RMSprop, as we would also have to scale the amount of shrinkage induced by the regularizer accordingly. This, in turn, would cause the fan-in of new units to become zero before they can learn anything useful.

We resolve this dilemma with a new training algorithm: Adaptive Radial-Angular Gradient Descent (*AdaRad*), shown in algorithm 1. Like in all the algorithms cited above, we begin each iteration by computing the gradient $G$ of the loss term over the current mini-batch (line 8). Then, for each $1 \le l \le L$ and $1 \le j \le d_l$, we decompose the sub-vector $[G_l(1, j), G_l(2, j), .., G_l(d_{l-1}, j)]$ into a component parallel to its corresponding fan-in $[W_l(1, j), W_l(2, j), .., W_l(d_{l-1}, j)]$ and a component orthogonal to it (line 11). Out of the two, we normalize only the orthogonal component (line 14) while the parallel component is left unaltered. Finally, the normalized orthogonal component of each sub-vector is added to its corresponding fan-in in radial-angular coordinates instead of cartesian coordinates (line 16). This ensures that it does not affect the length of the fan-in. Like the parallel component, we leave the induced shrinkage unaltered. Note that $\ell_2$ shrinkage acts only to shorten the length of each fan-in, but does not alter its direction. Hence, AdaRad with an $\ell_2$ regularizer applies a normalized shift to each fan-in that alters its direction but not its length (angular shift), as well as an un-normalized shift that includes shrinkage that alters the length of the fan-in but not its direction (radial shift, lines 15 and 17).

AdaRad has two step sizes: One for the radial and one for the angular shift, $\alpha_r$ and $\alpha_\phi$ respectively. This is desirable as they both control the behavior of the training algorithm in different ways. The radial step size controls how long it takes for the fan-in of a unit to be shrunk to zero, i.e. the time a unit has to learn something useful. On the other hand, the angular step size controls the general speed of learning and is tuned to achieve the quickest possible descent along the error surface.

Like RMSprop and unlike Adam, AdaRad does not make use of the principle of momentum. We have developed a variant called AdaRad-M that does. It is described in the appendix in section 7.3.

Using AdaRad over SGD incurs additional computational cost. However, that cost scales more gracefully than the cost of, for example, RMSprop. AdaRad normalizes at the granularity of fan-ins instead of the granularity of individual weights, so many of its operations scale only with the number of units and not with the number of weights in the network. In Table 1, we compare the costs of SGD,

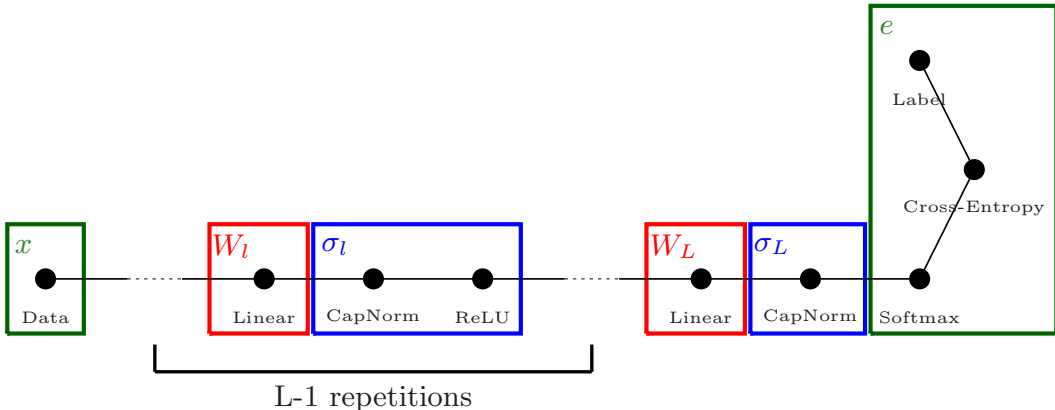

Figure 1: Architecture of the nonparametric networks used in the experiments. Activations flow rightward, gradients flow leftward. In color, we show how each element corresponds to our definition of a neural network in section 2. CapNorm does not fully fit our definition of nonlinearity as it requires information from multiple datapoints to compute its value. Hence, theorem 1 and proposition 1 do not technically apply. However, CapNorm is a benign operation that does not lead to problems in practice.

AdaRad and RMSprop. Further, RMSprop has a larger memory footprint than AdaRad. Compared to SGD, it requires an additional cache of size equal to the number of weights, whereas AdaRad only requires 2 additional caches of size equal to the number of units.

## 4 EXPERIMENTS

We evaluated our framework using the network architecture shown in Figure 1 with ReLU nonlinearities and CapNorm, and using AdaRad as the training algorithm. We used two hidden layers ($L = 3$) and started off with ten units in each hidden layer and each fan-in initialized randomly with expected length 1. We add one new unit with random fan-in of expected length 1 and zero fan-out to each layer every epoch. While this does not lead to fast convergence - we have to wait until tens or hundreds of units are added - we believe that growing nets from scratch is a good test case for investigating the robustness of our framework. After the validation error stopped improving, we ceased adding units, allowing all remaining redundant units to be eliminated. We set $\alpha_r = \frac{1}{50\lambda}$, as this allows each new unit $\approx 50$ epochs to train before being eliminated by shrinkage, assuming the length of the fan-in is not altered by the gradient of the loss term.

When training parametric networks, we replaced CapNorm with batch normalization, either with or without trainable free mean and variance parameters. We trained the network using one of the following algorithms: SGD, momentum, Nesterov momentum, RMSprop or Adam. Further experimental details can be found in the appendix in section 7.4.

### 4.1 PERFORMANCE

In this section, we investigate our two core questions: (A) Do nonparametric networks converge to a good size? (B) Do nonparametric networks achieve higher accuracy than parametric networks?

We evaluated our framework using three standard benchmark datasets - the *mnist* dataset, the *rectangles images* dataset and the *convex* dataset (Bergstra & Bengio, 2012). We started by training nonparametric networks. Through preliminary experiments, we determined a good starting angular step size for all datasets. We chose to start with $\alpha_\phi = 30$ and repeatedly divided $\alpha_\phi$ by 3 when the validation error stopped improving. By varying the random seed, we trained 10 nets each for several values of the regularization parameter $\lambda$ per dataset and then chose a typical representative from among those 10 trained nets. Results are shown in black in figure 2. Values of $\lambda$ are $3 * 10^{-3}$, $10^{-3}$ and $3 * 10^{-4}$ for *MNIST*, $3 * 10^{-5}$ and $10^{-6}$ for *rectangles images* and $10^{-5}$ and $10^{-8}$ for *convex*.

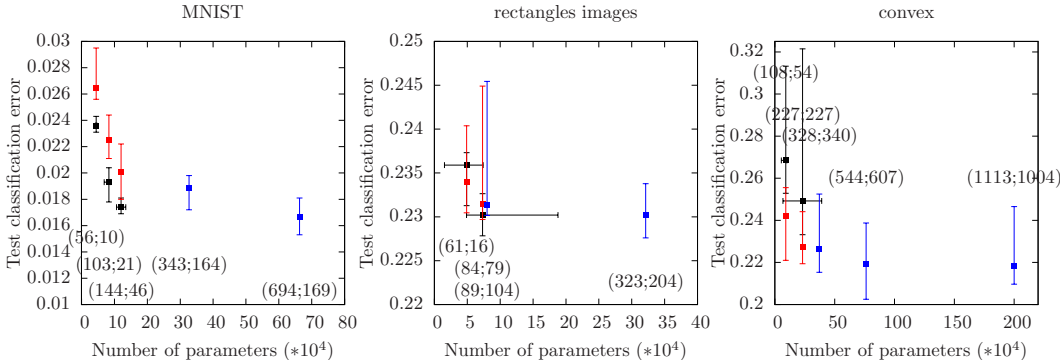

Figure 2: Test classification error of trained networks. Nonparametric networks are shown in black, parametric networks in red and blue. Error bars indicate the range over 10 random reruns of the same setting. For parametric networks, the square represents the median test error over those 10 runs. For nonparametric networks, the square represents the test error and size of a single representative run that was close to the median in both size and error. In brackets below or above each plotted point, we show the number of units in the two hidden layers.

Then, we trained parametric networks of the same size as the chosen representatives. The top performers after an exhaustive grid search are shown in red in figure 2. Finally, we conducted an exhaustive random search where we also varied the size of both hidden layers. The top performers are shown in blue in the same figure.

We obtain different results for the three datasets. For *mnist*, nonparametric networks substantially outperform parametric networks of the same size. The best nonparametric network is close in performance to the best parametric network, while being substantially smaller (144 first layer units versus 694). For *rectangles images*, nonparametric networks underperform parametric networks of the same size when $\lambda$ is large and outperform them when $\lambda$ is small. Here, the best nonparametric network has the globally best performance, as measured by the median test error over 10 random reruns, using substantially fewer parameters than the best parametric network.

While results for the first two datasets are very promising, nonparametric networks performed badly on the *convex* dataset. Parametric networks of the same size perform substantially better and also have a smaller range of performance across random reruns. Even if the model found by training nonparametric networks were re-trained as a parametric network, the apparent tendency of nonparametric networks to converge to relatively small sizes hurts us here as we would still miss out on a significant amount of performance.

We also conducted experiments with AdaRad-M, but found that performance was very similar to that of AdaRad. Hence, we omit the results. Similarly, we found no significant difference in performance between parametric networks trained with RMSprop and those trained with Adam.

## 4.2 ANALYSIS OF THE NONPARAMETRIC TRAINING PROCESS

In this section, we analyze in detail a single training run of a nonparametric network. We chose *mnist* as dataset, set $\lambda = 3 * 10^{-4}$ and lowered the angular step size to 10 as we did not use step size annealing. We trained for 1000 epochs while adding one unit to each hidden layer per epoch, then trained another 1000 epochs without adding new units. The final network had 193 units in the first hidden layer and 36 units in the second hidden layer. The results are shown in figure 3.

In part (A), we show the validation classification error. As a comparison, we trained two parametric networks with 193 and 36 hidden units for 1000 epochs, once using SGD and the same step size and $\lambda$ as the nonparametric network, and once using optimal settings (RMSprop, $\alpha = 300$, $\lambda = 0$). It is not suprising that the parametric networks reach a good accuracy level faster, as the nonparametric network must wait for its units to be added. Also, the parametric network benefits from an increased step size - in this case $\alpha = 300$. This was true throughout our experimental evaluation.

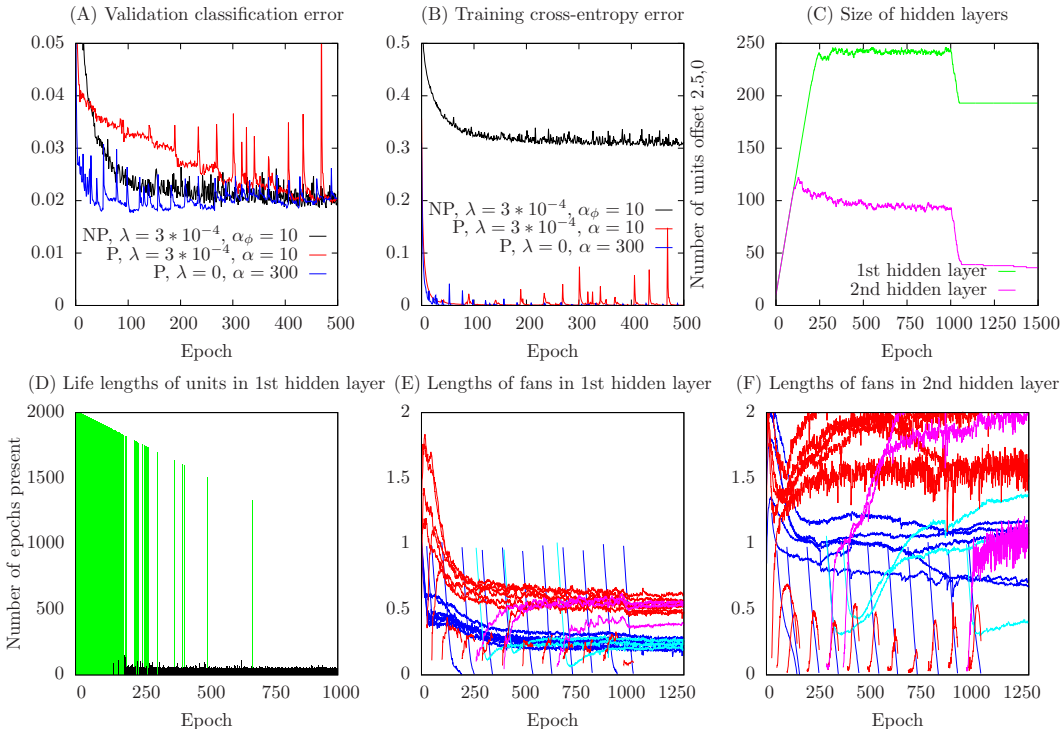

Figure 3: Detailed statistics of a nonparametric training run. See main text for details.

In (B), we show the training cross-entropy error for the same training runs. Interestingly, parametric networks reach an error very close to zero. In fact, the unregularized network reaches a value of $\approx 10^{-6}$ and the regularized network reaches a value of $\approx 10^{-4}$. Both made zero classification mistakes on the training set after training. In contrast, the nonparametric network did not have a near-zero training cross-entropy error. Towards the end of training, it still misclassified around 30 out of 50.000 training examples. However, this did not harm its performance on the validation or test set. In fact, the validation error of nonparametric networks tended to improve slowly for many epochs, whereas unregularized parametric networks (which were the best parametric networks when early stopping is used) tended to have a slightly increasing validation error in the long run.

In (C), we show the size of the two hidden layers during training. These curves are very typical of all training runs we examined. For the first $\approx 50$ epochs, no units are eliminated. This is because we chose $\alpha_r = \frac{1}{50\lambda}$, which guarantees that units that are added with a fan-in of length 1 take $\approx 50$ epochs to be eliminated, assuming no impact from the gradient of the loss term. If the layer requires a relatively large number of units, it will keep growing linearly for a while and then either plateau or shrink slightly. Once we no longer add units after 1000 epochs, both layers shrink linearly by $\approx 50$ units over $\approx 50$ iterations, as the units that were added roughly between epochs 950 and 1000 are eliminated in succession. Overall, this process shows the value of controlling $\alpha_\phi$ and $\alpha_r$ independently, as we can manage the "overhead" of extraneous units present during training while still ensuring an ideal speed of learning. In (D), we show the length of time individual units in the first hidden layer were present during training. On the x axis, we show the epoch during which a given unit was added. On the y axis, we show the number of epochs the unit was present. Green bars represent units that survived until the end, while black bars represent units that did not. As one might expect, units were more likely to survive the earlier they were added. Units that did not survive were eliminated in $\approx 50$ epochs. The same graph for the second hidden layer is shown in figure 4.

In (E) and (F), we show the lengths of fan-ins (blue) and fan-outs (red) of units in the hidden layers. For each layer, we depict the following units in dark colors: three randomly chosen units that were initially present as well as units that were added at epochs 0, 25, 50, 100, 200, 300, .., 1000. In addition, in light colors, we show three units that were added late but not eliminated. We see a

Table 2: Test classification error of various models trained on the *poker* dataset.

| Algorithm | $\lambda$ | Starting net size | Final net size | Error |
|---|---|---|---|---|
| Logistic regression (ours) | | | | 49.9% |
| Naive bayes (OpenML) | | | | 48.3% |
| Decision tree (OpenML) | | | | 26.8% |
| | $10^{-3}$ | 10-10-10-10 | 23-24-15-4 | 0.62% |
| | $10^{-5}$ | 10-10-10-10 | 94-135-105-35 | 0.022% |
| Nonparametric net | $10^{-6}$ | 10-10-10-10 | 210-251-224-104 | 0.001% |
| | $10^{-7}$ | 10-10-10-10 | 299-258-259-129 | 0% |
| | | 23-24-15-4 | unchanged | 0.20% |
| | | 94-135-105-35 | unchanged | 0.003% |
| Parametric net | | 210-251-224-104 | unchanged | 0.003% |
| | | 299-258-259-129 | unchanged | 0.002% |

consistent pattern for individual units. First, their length decreases linearly as the CapNorm layer filters the component of the gradient parallel to the fan-ins as long as the standard deviation of the pre-activations $\sigma$ exceeds 1. During this period, the unit learns something useful and so the fan-out increases in length. When finally $\sigma < 1$, the parallel component of the gradient starts to slow down the decay and, if the unit has become useful enough, reverses it. If the decay is not reversed, the unit is eliminated. If it is reversed, both fan-in and fan-out will attain a length comparable to those of well-established units.

From a global perspective, we notice that fan-ins in the first layer have lengths much less than 1. This is because first layer units encode primarily AND functions of highly correlated input features, meaning weights of small magnitude are sufficient to attain $\sigma = 1$. In contrast, lengths of fan-ins in the second layer are more chaotic. We found this is because $\sigma = 1$ is generally NOT attained in the second layer. In fact, the network compensated for lower activation values in the second layer by assigning fan-ins of stable lengths between 3.5 and 4.5 to the 10 output units. The network can assign these lengths dynamically without altering the output of the network because ReLU is self-similar, as described in section 3.1.

## 4.3 SCALABILITY

Finally, we wanted to verify whether nonparametric networks could be applied to a large dataset. We visited OpenML http://www.openml.org/, a website containing many datasets as well as the performance of various machine learning models applied to those datasets. We applied nonparametric networks to the largest classification dataset [1] on OpenML meeting our standards [2]. This was the *poker* dataset http://www.openml.org/d/354. It is a binary classification dataset with 1.025.010 datapoints and 14 features per datapoint. We had no prior information about this dataset. In general, we think that nonparametric networks are most useful in cases with no prior information and thus no possibility of choosing a good parametric model a priori.

We made the following changes to the experimental setup for *poker*: (i) we used 4 hidden layers instead of 2 (ii) we added a unit every tenth of an epoch instead of every epoch and (iii) we multiplied the radial step size by 10, i.e. $\alpha_r = \frac{1}{5\lambda}$. The latter two changes were made as *poker* is approximately one order of magnitude larger than *mnist*, and we wanted to approximately preserve the rate of unit addition and elimination per mini-batch. Those changes were made a priori and were not based on examining their performance.

After some exploration, we set the starting angular step size for nonparametric networks to 10. We trained nonparametric networks for various values of $\lambda$, obtaining nets of different sizes. We then trained parametric networks of those same sizes with RMSprop, where the step size was chosen by validation, independently for each network size.

---

[1] in terms of number of datapoints

[2] our standards were: at least 10 published classification accuracy values; no published classification accuracy values exceeding 95%; no extreme label imbalance

The results are shown in Table 2. Both parametric and nonparametric networks perform very well, achieving less than 1% test error even for small networks. The nonparametric networks had a higher error for larger values of $\lambda$ and a slightly lower error for smaller values of $\lambda$. In fact, the best nonparametric network made no mistake on the test set of 100.000 examples. For comparison, we show that linear models perform roughly as well as random guessing on *poker*. Also, the best result published on OpenML, achieved by a decision tree classifier, vastly underperforms our 4-hidden layer networks.

To achieve convergence, networks required many more mini-batches on *poker* than they did on the smaller datasets used in section 4.1. However, since units were added to the nonparametric networks at roughly the same rate per mini-batch, the time it took those networks to converge to a stable network size (as in Figure 3C) was a much smaller fraction of the overall training time under *poker* compared to the smaller datasets. Thus, the downside of increased training time as shown in Figure 3A incurred when networks are built gradually was ameliorated.

## 5 FURTHER BACKGROUND

Several strategies have been introduced to address the drawbacks of black-box model selection. Maclaurin et al. (2015) indeed calculate the gradient of the validation error after training with respect to certain hyperparameters, though their method only applies to specific networks trained with very specific algorithms. (Luketina et al., 2016) and (Larsen et al., 1998) train certain hyperparameters jointly with the network using second order information. Such methods are limited to continuous hyperparameters and are often applied specifically to regularization hyperparameters. Several papers try to speed up the global model search by estimating the validation error of trained networks without fully training them. Saxe et al. (2011) use the validation error with randomly initialized convolutional layers as a proxy. Klein et al. (2017) predict the validation error after training based on the progress made during the first few epochs.

Several papers have achieved increased performance by growing networks during training. Our main inspiration was Wei et al. (2016), who utilize a notion similar to our $f$-equivalence, though they enlarge their network in a somewhat ad-hoc way. The work of Chen et al. (2016) is similar, but focuses on convergence speed. Pandey & Dukkipati (2014) transform a trained small network into a larger network by multiplying weight matrices with large, random matrices.

The performance of a network of given size can be improved by injecting knowledge from other nets trained on the same task. Ba & Caruana (2014) use the predictions of a large network on a dataset to train a smaller network on those predictions, achieving an accuracy comparable to the large network. Hinton et al. (2015) compress the information stored in an ensemble of networks into a single network. Simonyan & Zisserman (2015) train very deep convolutional networks by initializing some layers with the trained layers of shallower networks. Romero et al. (2015) train deep, thin networks utilizing hints from wider, shallower networks.

Bayesian neural networks (e.g. McKay (1992), De Freitas (2003)) use a probabilistic prior instead of a regularizer to control the complexity of the network. Gaussian processes can been used to mimic "infinitely wide" neural networks (e.g. Williams (1997), Hazan & Jaakkola (2015)), thus eliminating the need to choose layer width and replacing it with the need to choose a kernel. Compared to these and other Bayesian approaches, we work within the popular feed-forward function optimization paradigm, which has advantages in terms of computational and algorithmic complexity.

Adding units to a network one at a time is an idea with a long history. Ash (1989) adds units to a single hidden layer, whereas Gallant (1986) builds up pyramid and tower structures and Fahlman & Lebiere (1990) effectively create a new layer for each new unit. While these papers provided inspiration to us, the methods they present for determining when to add a new unit requires training the network to convergence first, which is impractical in modern settings. We circumvent this problem by adding units agnostically and providing a mechanism for removing unnecessary units.

## 6 CONCLUSION

We introduced nonparametric neural networks - a simple, general framework for automatically adapting and choosing the size of a neural network during a single training run. We improved the performance of the trained nets beyond what is achieved by regular parametric networks of the same size and obtained results competitive with those of an exhaustive random search, for two of three datasets. While we believe there is room for performance improvement in several areas - e.g. unit initialization, unit addition schedule, additional regularization and starting network size - we see this paper as validation of the basic concept. We also proved the theoretical soundness of the framework.

In future work, we plan to extend our framework to include convolutional layers and to automatically choosing the depth of networks, as done by e.g. Wen et al. (2016). Part of our motivation to develop nonparametric networks was to control the layer size via a continuous parameter. We want to make use of this by tuning $\lambda$ during training, either by simple annealing or in a comprehensive framework such as the one introduced in Luketina et al. (2016). We want to use nonparametric networks to learn more complicated network topologies for e.g. semi-supervised or multi-task learning. Finally, we plan to investigate the possibility of sampling units with different nonlinearities and training an ever-growing network for lifelong learning.

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

## 7 APPENDIX

### 7.1 PROOF OF THEOREM 1

First, we restate the theorem formally.

**Theorem 1.** *For all*

- $L, d_0, d_L \in \mathbb{Z}_+$

- *finite datasets $D$ of points $(x, y)$ with $x \in \mathbb{R}^{d_0}$ and $y \in Y$ for some set $Y$*

- *sets of nonlinearities $\{\sigma_l : \mathbb{R} \to \mathbb{R}, 1 \le l \le L\}$ where each $\sigma_l$ fulfils the following conditions:*

  - *There exists a function $b_{1,l} : \mathbb{R}_{\ge 0} \to \mathbb{R}_{\ge 0}$ such that for all $S \in \mathbb{R}_{\ge 0}$, $-S \le s \le S$, we have $|\sigma_l(s)| \le b_{1,l}(S) * |s|$.*
  - *It is left- and right-differentiable everywhere.*
  - *There exists a function $b_{2,l} : \mathbb{R}_{\ge 0} \to \mathbb{R}_{\ge 0}$ such that for all $S \in \mathbb{R}_{\ge 0}$, $-S \le s \le S$, we have $|\sigma_l^{\leftarrow}(s)| \le b_{2,l}(S)$ and $|\sigma_l^{\rightarrow}(s)| \le b_{2,l}(S)$, where the superscripts indicate directional derivatives.*

- *error functions $e : (\mathbb{R}^{d_L} \times Y) \to \mathbb{R}$ that fulfils the following conditions:*

  - *It is non-negative everywhere.*
  - *It is differentiable with respect to its first argument everywhere.*
  - *There exists a function $b_3 : \mathbb{R}_{\ge 0} \to \mathbb{R}_{\ge 0}$ such that for all $S \in \mathbb{R}_{\ge 0}$, $v \in \mathbb{R}^{d_L}$ and $y \in Y$, we have $e(v, y) \le S \implies ||\frac{de(v,y)}{dv}||_\infty \le b_3(S)$*

- *$\lambda > 0$ and $1 \le p < \infty$*

- *$\Omega \in \{\Omega_{in}, \Omega_{out}\}$*

*we have that*

$$E(\mathbf{d}, \mathbf{W}) = \frac{1}{|D|} \sum_{(x,y) \in D} e(f(\mathbf{W}, x), y) + \Omega(\mathbf{W}, \lambda, p) \quad (4)$$

*attains a global minimum.*

Most commonly used nonlinearities are admissible under this theorem as long as $\sigma(0) = 0$, i.e. the sigmoid non-linearity is not admissible, but the $\tanh$ non-linearity is. Note that nonlinearities, away from zero, are allowed to grow at an almost arbitrary pace. For example, polynomial or even exponential nonlinearities are possible. Note that the first condition on nonlinearities is technically implied by the other two as long as $\sigma(0) = 0$, though we will not prove this.

The conditions for the error function cover the two most popular choices: cross-entropy coupled with softmax (as in Figure 1) - and the square of the $\ell_2$ distance.

We will prove this theorem through a sequence of lemmas. Throughout this process, all inputs to the main theorem are considered fixed and fulfilling their respective conditions.

**Lemma 1.** *Theorem 1 holds if $\mathbf{d}$ is fixed.*

I.e. in the parametric case, 4 attains a global minimum.

*Proof.* Let $d$ be fixed. Let $B = E(\mathbf{d}, \mathbf{0})$, where $\mathbf{0}$ is the value of $\mathbf{W}$ of dimensionality $\mathbf{d}$ where all individual weights are set to zero. Then let $\mathbf{W}_B$ be the space of all $\mathbf{W}$ of dimensionality $\mathbf{d}$ which have at least one individual weight with absolute value greater than $\frac{B}{\lambda}$. Clearly, $E(\mathbf{d}, \mathbf{W}) > B$ for all $\mathbf{W} \in \mathbf{W}_B$. Since $\mathbb{R}^{\mathbf{d}} \backslash \mathbf{W}_B$ is compact and $E$ is continuous, there exists a point $\mathbf{W}_{\min}$ that is a minimum of $E$ inside $\mathbb{R}^{\mathbf{d}} \backslash \mathbf{W}_B$. Further, $\mathbb{R}^{\mathbf{d}} \backslash \mathbf{W}_B$ contains at least one point, namely $\mathbf{0}$, for which

$E \leq B$, so a minimum within $\mathbb{R}^{\mathbf{d}} \backslash \mathbf{W}_B$ is indeed a global minimum, the existence of which was required. $\qquad \square$

Now, some definitions:

- We call a parameter value $(\mathbf{d}, \mathbf{W})$ a *local minimum* of $E$ iff it is a local minimum in its second component, $\mathbf{W}$.

- We call a local minimum of $E$ *$B$-locally minimal* for some $B \in \mathbb{R}$ iff the value of $E$ at that minimum does not exceed $B$.

- We call the *proper dimensionality* of $\mathbf{W}$ the dimensionality obtained when eliminating from $\mathbf{W}$ all units which have a zero fan-in or a zero fan-out or both.

- We call a parameter value $(\mathbf{d}, \mathbf{W})$ *proper* if $\mathbf{d}$ is the proper dimensionality of $\mathbf{W}$. We also call a local minimum with such a parameter value proper.

- Denote $(d_1, .., d_l)$ by $\mathbf{d}_{\leq l}$ and $(W_1, .., W_l)$ by $\mathbf{W}_{\leq l}$.

- $D = \{(x^{(0)}, y^{(0)}), (x^{(1)}, y^{(1)}), .., (x^{(N)}, y^{(N)})\}$

- We denote intermediate computations of the neural network $f(\mathbf{W}, x)$ as follows:

$$x_0 := x \tag{5}$$
$$z_l := x_{l-1} W_l \qquad 1 \leq l \leq L \tag{6}$$
$$x_l := \sigma_l.(z_l) \qquad 1 \leq l \leq L \tag{7}$$
$$f(\mathbf{W}, x) = x_L \tag{8}$$

- We denote the gradients of $e(f(\mathbf{W}, x), y)$, when they are defined, as follows:

$$g_l := \frac{de(f(\mathbf{W}, x), y)}{dx_l} \qquad 0 \leq l \leq L \tag{9}$$

$$h_l := \frac{de(f(\mathbf{W}, x), y)}{dz_l} \qquad 1 \leq l \leq L \tag{10}$$

$$G_l := \frac{de(f(\mathbf{W}, x), y)}{dW_l} \qquad 1 \leq l \leq L \tag{11}$$

- Vector and matrix indeces are written in brackets. For example, the $j$'th component of $z_l^{(n)}$ is denoted by $z_l^{(n)}(j)$.

- We denote by square brackets a vector and by its subscript the index the vector is over, e.g. $[v_i]_i$ is a vector over index $i$.

**Lemma 2.** *Under the conditions of theorem 1 and the additional condition that the $\sigma_l$ are differentiable everywhere, if $\Omega$ is the fan-in regularizer, then for all $B$, the set of values of $\mathbf{d}$ for which there exist proper $B$-local minima is bounded.*

**Lemma 3.** *Under the conditions of theorem 1 and the additional condition that the $\sigma_l$ are differentiable everywhere, if $\Omega$ is the fan-out regularizer, then for all $B$, the set of values of $\mathbf{d}$ for which there exist proper $B$-local minima is bounded.*

Lemmas 2 and 3 are the core segments of the overall proof. Here we show that that very large nets have no "good" local minima.

*Proof of lemma 2.* Throughout this proof, we consider $B$ fixed.

*Claim 1a*: There exist constants $B_{x,l}$, $0 \leq l \leq L$, such that at all proper $B$-local minima, for all $1 \leq n \leq N$, for all $0 \leq l \leq L$, we have $||x_l^{(n)}||_1 \leq B_{x,l}$.

*Claim 1b*: There exist constants $B_{z,l}$, $1 \leq l \leq L$, such that at all proper $B$-local minima, for all $1 \leq n \leq N$, for all $1 \leq l \leq L$, we have $||z_l^{(n)}||_1 \leq B_{z,l}$.

*Claim 1c*: There exist constants $B_{d\sigma,l}$, $1 \leq l \leq L$, such that at all proper $B$-local minima, for all $1 \leq n \leq N$, for all $1 \leq l \leq L$, for all $1 \leq j \leq d_l$, we have $|\sigma'(z_l^{(n)}(j))| \leq B_{d\sigma,l}$.

First, we notice that it is sufficient to prove the bounds exist for a specific datapoint. The uniform bound across all datapoints is then simply the maximum of the individual bounds. Denote by $(x, y)$ an arbitrary fixed datapoint throughout the proof of the above claims. Also, notice that the claims are trivially true if there are no proper $B$-local minima. Hence, throughout the proof of the claims, we assume there exists at least one such minimum.

We will prove the claims jointly by induction. The order of the induction follows the order of computation of the neural network. Our starting case will be $x_0$, followed by $z_1$, $f_1'$ and $x_1$ etc.

The starting case is obvious as $x_0 = x$ is fixed and does not depend on the parameter $(\mathbf{d}, \mathbf{W})$. Hence we can choose $B_{x,0} = ||x||_1$.

Now assume we have $B_{x,l-1}$ such that $\sup_{(\mathbf{d},\mathbf{W})\text{proper }B\text{-locally minimal}} ||x_{l-1}||_1 \leq B_{x,l-1}$. Then:

$$\sup_{(\mathbf{d},\mathbf{W})\text{proper }B\text{-locally minimal}} ||z_l||_1 \tag{12}$$

$$= \sup_{(\mathbf{d},\mathbf{W})\text{proper }B\text{-locally minimal}} ||x_{l-1}W_l||_1 \tag{13}$$

$$\leq \sup_{(\mathbf{d},\mathbf{W}),||x_{l-1}||_1 \leq B_{x,l-1}, \lambda \sum_{j=1}^{d_l} ||[W_l(i,j)]_i||_p \leq B} ||x_{l-1}W_l||_1 \tag{14}$$

$$= \sup_{\mathbf{d}_{\leq l}} (\sup_{W_l, \lambda \sum_{j=1}^{d_l} ||[W_l(i,j)]_i||_p \leq B} (\sup_{\mathbf{W}_{<l}, ||x_{l-1}||_1 \leq B_{x,l-1}} ||x_{l-1}W_l||_1)) \tag{15}$$

$$\leq \sup_{\mathbf{d}_{\leq l}} (\sup_{W_l, \lambda \sum_{j=1}^{d_l} ||[W_l(i,j)]_i||_p \leq B} (\sup_{u, \dim(u)=d_{l-1}, ||u||_1 \leq B_{x,l-1}} ||u^T W_l||_1)) \tag{16}$$

$$= \sup_{\mathbf{d}_{\leq l}} (\sup_{W_l, \lambda \sum_{j=1}^{d_l} ||[W_l(i,j)]_i||_p \leq B} (\sup_{u, \dim(u)=d_{l-1}, ||u||_1 \leq B_{x,l-1}} \sum_{j=1}^{d_l} |u^T [W_l(i,j)]_i|)) \tag{17}$$

$$= \sup_{\mathbf{d}_{\leq l}} (\sup_{c_j \geq 0, \sum_{j=1}^{d_l} c_j \leq \frac{B}{\lambda}} (\sup_{W_l, ||[W_l(i,j)]_i||_p = c_j} (\sup_{u, \dim(u)=d_{l-1}, ||u||_1 \leq B_{x,l-1}} \sum_{j=1}^{d_l} |u^T [W_l(i,j)]_i|))) \tag{18}$$

$$\leq \sup_{\mathbf{d}_{\leq l}} (\sup_{c_j \geq 0, \sum_{j=1}^{d_l} c_j \leq \frac{B}{\lambda}} \sum_{j=1}^{d_l} (\sup_{W_l, ||[W_l(i,j)]_i||_p = c_j} (\sup_{u, \dim(u)=d_{l-1}, ||u||_1 \leq B_{x,l-1}} |u^T [W_l(i,j)]_i|))) \tag{19}$$

$$= \sup_{\mathbf{d}_{\leq l}} (\sup_{c_j \geq 0, \sum_{j=1}^{d_l} c_j \leq \frac{B}{\lambda}} \sum_{j=1}^{d_l} (\sup_{v, \dim(v)=d_{l-1}, ||v||_p = c_j} (\sup_{u, \dim(u)=d_{l-1}, ||u||_1 \leq B_{x,l-1}} |u^T v|))) \tag{20}$$

$$\leq \sup_{\mathbf{d}_{\leq l}} (\sup_{c_j \geq 0, \sum_{j=1}^{d_l} c_j \leq \frac{B}{\lambda}} \sum_{j=1}^{d_l} (\sup_{v, \dim(v)=d_{l-1}, ||v||_\infty \leq c_j, u, \dim(u)=d_{l-1}, ||u||_1 \leq B_{x,l-1}} |u^T v|)) \tag{21}$$

$$\leq \sup_{\mathbf{d}_{\leq l}} (\sup_{c_j \geq 0, \sum_{j=1}^{d_l} c_j \leq \frac{B}{\lambda}} \sum_{j=1}^{d_l} c_j B_{x,l-1}) \tag{22}$$

$$\leq \sup_{\mathbf{d}_{\leq l}} \frac{B B_{x,l-1}}{\lambda} \tag{23}$$

$$= \frac{B B_{x,l-1}}{\lambda} \tag{24}$$

A line-by-line explanation of the above is as follows:

 13 Replacing $z_l$ by its definition.

14 Relaxing the conditions on $(\mathbf{d}, \mathbf{W})$ by replacing proper $B$-local minimality by two conditions that proper $B$-local minimality implies. The first condition is the induction hypothesis. The second condition follows because $E \leq B$ and so specifically $\Omega_{in}(\mathbf{W}) \leq B$ and so specifically $\Omega_{in}(W_l) \leq B$

15 Breaking up the supremum into three stages. We drop components of $\mathbf{d}$ and $\mathbf{W}$ that are immaterial to the value of the objective of the supremum.

16 We further relax the innermost sup by no longer requiring that $x_{l-1}$ be the intermediate output of some neural network but an arbitrary vector of fixed size and limited length. $\mathbf{W}_{<l}$ then becomes immaterial.

17 Replacing the $\ell_1$ norm by its definition.

18 We fix the length of each fan-in in the second sup and add an additional sup over these lengths.

19 Jensen's inequality

20 Simplifying the notation by replacing rows of $W_l$ by vector $v$.

21 Relaxing the conditions on $v$.

22 Using an elementary property of norms.

23 obvious

24 obvious

And therefore, we may choose $B_{z,l} = \frac{B B_{x,l-1}}{\lambda}$ as required.

Now consider the other inductive steps. Assuming we have a valid $B_{z,l}$, we have at all proper $B$-local minima $|\sigma_l'(z_l(j))| \leq b_{2,l}(B_{z,l})$ because $|z_l(j)| \leq ||z_l||_1 \leq B_{z,l}$ and hence we can choose $B_{d\sigma,l} = b_{2,l}(B_{z,l})$ as required. Finally, at all proper $B$-local minima, $||x_l||_1 = ||\sigma_l.(z_l)||_1 = \sum_{j=1}^{d_l} |\sigma_l(z_l(j))| \leq \sum_{j=1}^{d_l} b_{1,l}(B_{z,l})|z_l(j)| = b_{1,l}(B_{z,l})||z_l||_1 \leq b_{1,l}(B_{z,l})B_{z,l}$ and so we can choose $B_{x,l} = b_{1,l}(B_{z,l})B_{z,l}$. This completes the proof of claims 1(a)-(c).

*Claim 2a*: There exist constants $B_{g,l}$, $0 \leq l \leq L$, such that at all proper $B$-local minima, for all $1 \leq n \leq N$, for all $0 \leq l \leq L$, for all $1 \leq i \leq d_l$, we have $|g_l^{(n)}(i)| \leq B_{g,l}$.

*Claim 2b*: There exist constants $B_{h,l}$, $1 \leq l \leq L$, such that at all proper $B$-local minima, for all $1 \leq n \leq N$, for all $1 \leq l \leq L$, for all $1 \leq i \leq d_l$, we have $|h_l^{(n)}(i)| \leq B_{h,l}$.

Again, we can restrict our attention to a single datapoint and again, we will prove these claims by induction, but going backwards along the flow of the gradient. The starting case is $g_L$. At all proper $B$-local minima, $E \leq B$, so specifically $e(x_L, y) \leq B$ and therefore we have $||g_L||_\infty = ||\frac{de(x_L, y)}{dx_L}||_\infty \leq b_3(B)$ and so specifically $|g_L(i)| \leq b_3(B)$ and so we can choose $B_{g,L} = b_3(B)$ as required.

Now we assume we have a valid $B_{g,l}$. At all proper $B$-local minima we have $|h_l(i)| = |\frac{de}{dz_l(i)}| = |\frac{de}{dx_l(i)}\frac{dx_l(i)}{dz_l(i)}| = |g_l(i)||\sigma'(z_l(i))| \leq B_{g,l}B_{d\sigma,l}$. Therefore we can choose $B_{h,l} = B_{g,l}B_{d\sigma,l}$ as required.

Finally, assume we have $B_{h,l}$. Then we have $|g_{l-1}(i)| = |\frac{de}{dx_{l-1}(i)}| = |\sum_{j=1}^{d_l} \frac{de}{dz_l(j)}\frac{dz_l(j)}{dx_l(i)}| = |\sum_{j=1}^{d_l} h_l(j)W_l(i,j)| \leq \sum_{j=1}^{d_l} |h_l(j)||W_l(i,j)| \leq B_{h,l}\sum_{j=1}^{d_l}|W_l(i,j)| \leq B_{h,l}\sum_{j=1}^{d_l}||[W_l(i,j)]_i||_p \leq B_{h,l}B$, so we can choose $B_{g,l-1} = B_{h,l}B$ as required.

*Claim 3*: There is a constant $B_2$, such that at all proper $B$-local minima, for all $1 \leq l \leq L$, for all $1 \leq j \leq d_l$, we have $\sum_{n=1}^{N} |g_l^{(n)}(j)| \geq B_2$.

At any proper parameter value, all fan-ins are non-zero. Therefore $\Omega$ is differentiable with respect to $\mathbf{W}$, and therefore $E$ is differentiable with respect to $\mathbf{W}$. Hence, at any proper $B$-

local minimum, we have $\nabla_{\mathbf{W}} E = 0$, so in particular for any $l, j$ we have $\frac{dE}{d[W_l(i,j)]_i} = 0$, so $\frac{1}{N} \sum_{n=1}^{N} \frac{de}{d[W_l(i,j)]_i} = -\frac{d\Omega}{d[W_l(i,j)]_i}$ and so specifically $||\frac{1}{N} \sum_{n=1}^{N} \frac{de}{d[W_l(i,j)]_i}||_{\frac{p}{p-1}} = ||\frac{d\Omega}{d[W_l(i,j)]_i}||_{\frac{p}{p-1}}$, where $\frac{p}{p-1}$ can take the value $\infty$. Further analyzing the right hand side, we have $||\frac{d\Omega}{d[W_l(i,j)]_i}||_{\frac{p}{p-1}} = ||\frac{d(\lambda||[W_l(i,j)]_i||_p)}{d[W_l(i,j)]_i}||_{\frac{p}{p-1}} = \lambda$. Therefore, at all proper $B$-local minima we have:

$$\lambda \tag{25}$$

$$= \frac{1}{N} || \sum_n \frac{de}{d[W_l(i,j)]_i} ||_{\frac{p}{p-1}} \tag{26}$$

$$= \frac{1}{N} || \sum_n \frac{de}{dx_l(j)} \frac{dx_l(j)}{d[W_l(i,j)]_i} ||_{\frac{p}{p-1}} \tag{27}$$

$$= \frac{1}{N} || \sum_n g_l(j) \frac{dx_l(j)}{dz_l(j)} \frac{dz_l(j)}{d[W_l(i,j)]_i} ||_{\frac{p}{p-1}} \tag{28}$$

$$= \frac{1}{N} || \sum_n g_l(j) \sigma'_l(z_l(j)) x_{l-1} ||_{\frac{p}{p-1}} \tag{29}$$

$$\leq \frac{1}{N} || \sum_n g_l(j) \sigma'_l(z_l(j)) x_{l-1} ||_1 \tag{30}$$

$$= \frac{1}{N} \sum_{i=1}^{d_{l-1}} | \sum_n g_l(j) \sigma'_l(z_l(j)) x_{l-1}(i) | \tag{31}$$

$$\leq \frac{1}{N} \sum_{i=1}^{d_{l-1}} \sum_n |g_l(j)| |\sigma'_l(z_l(j))| |x_{l-1}(i)| \tag{32}$$

$$= \frac{1}{N} \sum_n |g_l(j)| |\sigma'_l(z_l(j))| ||x_{l-1}||_1 \tag{33}$$

$$\leq \frac{1}{N} \sum_n |g_l(j)| B_{d\sigma,l} B_{x,l-1} \tag{34}$$

Hence, we can choose $B_2 = \max_{1 \leq l \leq L} \frac{\lambda N}{B_{d\sigma,l} B_{x,l-1}}$ as required. Note that in the above equations, we have omitted all $^{(n)}$ superscripts for brevity.

*Claim 4*: There exist constants $D_l$, $1 \leq l \leq L$ such that at all proper $B$-local minima, for all $1 \leq l \leq L$, we have $d_l \leq D_l$.

Note that this claim is tantamount to proving the hypothesis of the Lemma.

We will prove this by induction going backwards. Since $d_L$ is fixed, we can simply pick $D_L = d_L$. Now assume we have a valid $D_{l+1}$. Then at all proper $B$-local minima we have

$$d_l B_2 \tag{35}$$

$$\leq \sum_{n=1}^{N} \sum_{i=1}^{d_l} |g_l^{(n)}(i)| \tag{36}$$

$$= \sum_{n=1}^{N} \sum_{i=1}^{d_l} | \sum_{j=1}^{d_{l+1}} W_{l+1}(i,j) h_{l+1}^{(n)}(j) | \tag{37}$$

$$\leq \sum_{n=1}^{N} \sum_{i=1}^{d_l} \sum_{j=1}^{d_{l+1}} |W_{l+1}(i,j) h_{l+1}^{(n)}(j) | \tag{38}$$

Therefore, by the box principle, there exists an $n'$ and $j'$ such that $\sum_{i=1}^{d_l} |W_{l+1}(i,j')h_{l+1}^{(n')}(j')| \geq \frac{d_l B_2}{d_{l+1} N}$. So further, we have

$$\frac{d_l B_2}{d_{l+1} N} \tag{39}$$

$$\leq \sum_{i=1}^{d_l} |W_{l+1}(i,j')h_{l+1}^{(n')}(j')| \tag{40}$$

$$\leq |h_{l+1}^{(n')}(j')| \sum_{i=1}^{d_l} |W_{l+1}(i,j')| \tag{41}$$

$$\leq B_{h,l+1} ||[W_{l+1}(i,j')]_i||_1 \tag{42}$$

$$\leq B_{h,l+1} d_l^{\frac{p-1}{p}} ||[W_{l+1}(i,j')]_i||_p \tag{43}$$

$$\leq B_{h,l+1} d_l^{\frac{p-1}{p}} B \tag{44}$$

And therefore, $d_l \leq (\frac{BB_{h,l+1}Nd_{l+1}}{B_2})^p$ and so we can choose $D_l = (\frac{BB_{h,l+1}ND_{l+1}}{B_2})^p$, which completes the proof.

$\square$

For brevity, we will only give a summary of the proof of lemma 3, as it is very similar to the proof of lemma 2.

*Sketch of proof of lemma 3.* As in the previous proof, we consider $B$ fixed.

*Claim 1a*: There exist constants $B_{x,l}$, $0 \leq l \leq L$, such that at all proper $B$-local minima, for all $1 \leq n \leq N$, for all $0 \leq l \leq L$, for all $1 \leq j \leq d_l$, we have $|x_l^{(n)}(j)| \leq B_{x,l}$.

*Claim 1b*: There exist constants $B_{z,l}$, $1 \leq l \leq L$, such that at all proper $B$-local minima, for all $1 \leq n \leq N$, for all $1 \leq l \leq L$, for all $1 \leq j \leq d_l$, we have $|z_l^{(n)}(j)| \leq B_{z,l}$.

*Claim 1c*: There exist constants $B_{d\sigma,l}$, $1 \leq l \leq L$, such that at all proper $B$-local minima, for all $1 \leq n \leq N$, for all $1 \leq l \leq L$, for all $1 \leq j \leq d_l$, we have $|\sigma'(z_l^{(n)}(j))| \leq B_{d\sigma,l}$.

As in the previous proof, we proceed by induction along the order of feed-forward execution of the neural network. However, we use the arguments we used for Claims 2(a)-(b) in the previous proof. This is because the fan-out regularizer "appears" like a fan-in regularizer when the direction of signal flow is reversed.

*Claim 2a*: There exist constants $B_{g,l}$, $0 \leq l \leq L$, such that at all proper $B$-local minima, for all $1 \leq n \leq N$, for all $0 \leq l \leq L$, we have $||g_l^{(n)}||_1 \leq B_{g,l}$.

*Claim 2b*: There exist constants $B_{h,l}$, $1 \leq l \leq L$, such that at all proper $B$-local minima, for all $1 \leq n \leq N$, for all $1 \leq l \leq L$, we have $||h_l^{(n)}||_1 \leq B_{h,l}$.

As in the previous proof, we proceed by induction along the flow of the gradient. However, we use the arguments we used for Claims 1(a)-(c) in the previous proof.

*Claim 3*: There is a constant $B_2$, such that at all proper $B$-local minima, for all $0 \leq l \leq L - 1$, for all $1 \leq j \leq d_l$, we have $\sum_{n=1}^{N} |x_l^{(n)}(j)| \geq B_2$.

*Claim 4*: There exist constants $D_l$, $0 \leq l \leq L - 1$ such that at all proper $B$-local minima, for all $0 \leq l \leq L - 1$, we have $d_l \leq D_l$.

The arguments mirror those of Claim 3 and 4 from the previous proof, but with the role of activation and gradient reversed. Also, here, Claim 4 is proved by induction along the order of feed-forward execution.

$\square$

**Lemma 4.** *Under the conditions of theorem 1, for all B, the set of values of $\mathbf{d}$ for which there exist proper B-local minima is bounded.*

This is the stronger version of the previous lemmas where we only use directional differentiability of the $\sigma_l$ instead of actual differentiability. The proof is a rather tedious extension of the previous two proofs and not very instructive, which is why we broke out the differential case as its own lemmas. Following this lemma, we immediately prove the main theorem.

*Proof.* We will describe how to amend the proof of lemma 2. The proof of lemma 3 can be amended similarly.

First, we define a *signature* $\mathbf{S}$ with dimensionality $\mathbf{d}$ as a binary sequence of vectors. Then, for all $\mathbf{d}$, $x' \in \mathbb{R}^{d_0}$, $\mathbf{W}'$ of dimensionality $\mathbf{d}$ and $\mathbf{S}$ of dimensionality $\mathbf{d}$, we define a *linearized neural network that is linearized at point $x'$ with weights $\mathbf{W}'$ and signature $\mathbf{S}$* $f^{[\mathbf{S},\mathbf{W}',x']}$ as follows: First, obtain $x'_l$ and $z'_l$ for $1 \le l \le L$ by evaluating $f(\mathbf{W}', x')$ as usual. Then, for each $\sigma_l$ used in $f$, define a vector of functions $\sigma_l^{[\mathbf{S},\mathbf{W}',x']}$, where each element is a linear function with $\sigma_l^{[\mathbf{S},\mathbf{W}',x']}(j)(z'_l(j)) = \sigma_l(z'_l(j))$ and with $\frac{d\sigma_l^{[\mathbf{S},\mathbf{W}',x']}(j)(s)}{ds} = \sigma_l^{\leftarrow}(z'_l(j))$ if $S_l(j) = 0$ and with $\frac{d\sigma_l^{[\mathbf{S},\mathbf{W}',x']}(j)(s)}{ds} = \sigma_l^{\rightarrow}(z'_l(j))$ if $S_l(j) = 1$. Finally, we obtain $f^{[\mathbf{S},\mathbf{W}',x']}$ from $f$ by, for each $1 \le l \le L$, replacing $\sigma_l$ that is applied elementwise with $\sigma_l^{[\mathbf{S},\mathbf{W}',x']}$ where each component is applied to the respective component of $z_l$.

In plain language, we linearize a neural network at a point by evaluating it at that point and replacing each nonlinearity by a straight line as indicated by the value and directional derivative of that nonlinearity wherever it is evaluated, where the direction of the derivative used is governed by $\mathbf{S}$.

Similarly, define a *partially linearized neural network* $f^{[\mathbf{S}_{\ge l},\mathbf{W}',x']}$ in the same fashion, except only layers $l$ and above are linearized. Finally, define a partially linearized neural network $f^{[\mathbf{S}_{\ge l,i},\mathbf{W}',x']}$ in the same fashion, except only layers $l$ and above as well as unit $i$ in layer $l-1$ are linearized.

$e$ is composed of functions that are differentiable or directionally differentiable with respect to $\mathbf{W}$, so $e$ itself is directionally differentiable with respect to $\mathbf{W}$. Specifically, let us analyze the directional derivative of $e$ with respect to some perturbation of $W_{L-1}$.

$$\nabla_{\delta W_{L-1}} e(f(\mathbf{W}, x), y) \tag{45}$$

$$= \nabla_{\delta W_{L-1}} e(\sigma_L.(\sigma_{L-1}.(x_{L-2}W_{L-1})W_L), y) \tag{46}$$

$$= \nabla_{\delta x_L = \nabla_{\delta W_{L-1}} \sigma_L.(\sigma_{L-1}.(x_{L-2}W_{L-1})W_L)} e(x_L, y) \tag{47}$$

$$= \frac{de}{dx_L} \nabla_{\delta W_{L-1}} (\sigma_L.(\sigma_{L-1}.(x_{L-2}W_{L-1})W_L))^T \tag{48}$$

$$= \sum_{j=1}^{d_L} \frac{de}{dx_L}(j) \nabla_{\delta W_{L-1}} (\sigma_L.(\sigma_{L-1}.(x_{L-2}W_{L-1})W_L))(j) \tag{49}$$

$$= \sum_{j=1}^{d_L} \frac{de}{dx_L}(j) \nabla_{\delta z_L = \nabla_{\delta W_{L-1}}[\sigma_{L-1}.(x_{L-2}W_{L-1})W_L]} (\sigma_L.(z_L))(j) \tag{50}$$

$$= \sum_{j=1}^{d_L} \frac{de}{dx_L}(j) \sigma_L^*(z_L(j)) \nabla_{\delta W_{L-1}} (\sigma_{L-1}.(x_{L-2}W_{L-1})W_L)(j) \tag{51}$$

$$= \sum_{j=1}^{d_L} \frac{de}{dx_L}(j) \sigma_L^*(z_L(j)) \sum_{i=1}^{d_{L-1}} W_L(i,j) \nabla_{\delta W_{L-1}} (\sigma_{L-1}.(x_{L-2}W_{L-1}))(i) \tag{52}$$

$$= (\frac{de}{dx_L}.*\sigma_L^*.(z_L)) W_L^T \nabla_{\delta W_{L-1}} (\sigma_{L-1}.(x_{L-2}W_{L-1}))^T \tag{53}$$

$$= \sum_{j=1}^{d_{L-1}} ((\frac{de}{dx_L}. * \sigma_L^*.(z_L))W_L^T)(j)\sigma_{L-1}^*(z_{L-1}(j))\nabla_{\delta W_{L-1}}(x_{L-2}W_{L-1})(j) \tag{54}$$

$$= \sum_{j=1}^{d_{L-1}} ((\frac{de}{dx_L}. * \sigma_L^*.(z_L))W_L^T)(j)\sigma_{L-1}^*(z_{L-1}(j)) \sum_{i=1}^{d_{L-2}} x_{L-2}(i)\nabla_{\delta W_{L-1}}W_{L-1}(i,j) \tag{55}$$

$$= \sum_{j=1}^{d_{L-1}} ((\frac{de}{dx_L}. * \sigma_L^*.(z_L))W_L^T)(j)\sigma_{L-1}^*(z_{L-1}(j)) \sum_{i=1}^{d_{L-2}} x_{L-2}(i)\delta W_{L-1}(i,j) \tag{56}$$

$$= (((( \frac{de}{dx_L}. * \sigma_L^*.(z_L))W_L^T). * \sigma_{L-1}^*.(z_{L-1}))^T x_{L-2}).\delta W_{L-1}(i,j) \tag{57}$$

Here, a $*$ superscript is a "wildcard" that can stand for a left or a right derivative. When combined with the .() elementwise operation, it can mean a different derivative (left or right) for each element.

We use the chain rule for directional derivatives (lines 47, 50 and 54), the linearity of the directional derivative (lines 52 and 55), the fact that the directional derivative of a differentiable function is the dot product of its gradient with the perturbation (line 48), and the fact that the directional derivative of a left- and right-differentiable scalar function is either the product of its left derivative with the perturbation or the product of its right derivative with the perturbation (lines 51 and 54).

We notice that the final expression in line 57 is the same expression we would obtain if the $\sigma_l$ were differentiable, except with a $*$ instead of a $'$ superscript. Now the linearized neural networks come into play. We can choose a signature $S$ that matches the wildcards in the above directional derivative and get $\nabla_{\delta W_{L-1}} e(f(\mathbf{W}, x), y) = \frac{de(f^{[\mathbf{S}, \mathbf{W}, x]}(\mathbf{W}, x), y)}{dW_{L-1}}.\delta W_{L-1}$, because the forward evaluation of $f^{[\mathbf{S}, \mathbf{W}, x]}$ and $f$ are identical at $x$ and the backward evaluation picks out the correct left and right derivatives. In fact, it is sufficient to choose a partially linearized network with signature $\mathbf{S}_{\geq L-1}$ to achieve the above identity.

So far, we have investigated the directional derivative with respect to $\delta W_{L-1}$. However, the same arguments hold for all $1 \leq l \leq L$. We can expand $\nabla_{\delta W_l} e(f(\mathbf{W}, x), y)$ in the same way, except we repeat the transformation from line 48 to line 53 $L - l$ times. Hence, we have what we will call claim 0.

*Claim 0*: For all $(\mathbf{d}, \mathbf{W})$, for all $x \in \mathbb{R}^{d_0}$, for all $1 \leq l \leq L$, for all $\delta W_l$, we can choose a signature $\mathbf{S}$ or a partial signature $\mathbf{S}_{\geq l}$, such that $\nabla_{\delta W_l} e(f(\mathbf{W}, x), y) = \frac{de(f^{[\mathbf{S}, \mathbf{W}, x]}(\mathbf{W}, x), y)}{dW_l}.\delta W_l$.

Now, we refer back to the proof of lemma 2. Claims 1(a)-(c) hold as before, except in Claim 1(c) we replace $|\sigma'(z_l^{(n)}(j))| \leq B_{d\sigma, l}$ with $|\sigma^\leftarrow(z_l^{(n)}(j))| \leq B_{d\sigma, l}$ and $|\sigma^\rightarrow(z_l^{(n)}(j))| \leq B_{d\sigma, l}$. Further, note that claims 1(a)-(c) also hold for neural networks that are linearized at the respective $B$-local minimum and datapoint at which they are evaluated.

Claims 2(a)-(b) are changed as follows:

*Claim 2a*: There exist constants $B_{g, l}$, $0 \leq l \leq L$, such that at all proper $B$-local minima, for all $1 \leq n \leq N$, for all $0 \leq l \leq L$, for all $1 \leq i \leq d_l$, for all signatures $\mathbf{S}$ or partial signatures $\mathbf{S}_{\geq l+1}$ of matching dimensionality we have $|g_l^{(n)[\mathbf{S}, \mathbf{W}, x^{(n)}]}(i)| \leq B_{g, l}$.

*Claim 2b*: There exist constants $B_{h, l}$, $1 \leq l \leq L$, such that at all proper $B$-local minima, for all $1 \leq n \leq N$, for all $1 \leq l \leq L$, for all $1 \leq i \leq d_l$, for all signatures $\mathbf{S}$ or partial signatures $\mathbf{S}_{\geq l+1, i}$ of matching dimensionality we have $|h_l^{(n)[\mathbf{S}, \mathbf{W}, x^{(n)}]}(i)| \leq B_{h, l}$.

The proof is as before, where derivatives of the $\sigma_l$ are again replaced by a left or right derivative as indicated by the signature.

Claim 3 and its proof change somewhat.

*Claim 3*: There exists a constant $B_2$, such that at all proper $B$-local minima, for all $1 \leq l \leq L$, for all $1 \leq j \leq d_l$, we have $\sum_{n=1}^N \sum_{\mathbf{S}_{\geq l+1}} |g_l^{(n)[\mathbf{S}_{\geq l+1}, \mathbf{W}, x^{(n)}]}(j)| \geq B_2$, where $\sum_{\mathbf{S}_{\geq l+1}}$ is the sum over all partial signatures.

At all proper $B$-local minima, we have for all $l$, $i$ and $j$:

$$0 \tag{58}$$

$$\leq \quad \nabla_{\delta W_l(i,j)} E \tag{59}$$

$$= \quad \nabla_{\delta W_l(i,j)}(\frac{1}{N}\sum_{n=1}^{N} e(f(\mathbf{W}, x^{(n)}), y^{(n)}) + \Omega(\mathbf{W})) \tag{60}$$

$$= \quad \frac{1}{N}\sum_{n=1}^{N} \nabla_{\delta W_l(i,j)} e(f(\mathbf{W}, x^{(n)}), y^{(n)}) + \nabla_{\delta W_l(i,j)}\Omega(W_l) \tag{61}$$

$$= \quad \frac{1}{N}\sum_{n=1}^{N} \frac{de(f^{[\mathbf{S}_{\geq l}^{i,j,n}, \mathbf{W}, x^{(n)}]}(\mathbf{W}, x^{(n)}), y^{(n)})}{dW_l(i,j)}\delta W(i,j) + \frac{d\Omega(W_l)}{dW_l(i,j)}\delta W(i,j) \tag{62}$$

Here, $\nabla_{\delta W_l(i,j)}$ stands for the directional derivative with respect to a change in the scalar value $W_l(i,j)$. Because this is a special case of a directional derivative with respect to $\delta W_l$, we can use Claim 0 to obtain line 62. Note that to use Claim 0, we have to choose a different partial signature for each value of $i$, $j$, and $n$, which we indicate by superscript. Specifically, we now choose $\delta W_l(i,j) = -1$ and corresponding signatures. Then, for all $l$, $i$ and $j$ we have $0 \leq \frac{1}{N}\sum_{n=1}^{N} \frac{de(f^{[\mathbf{S}_{\geq l}^{i,j,n}, \mathbf{W}, x^{(n)}]}(\mathbf{W}, x^{(n)}), y^{(n)})}{dW_l(i,j)}(-1) + \frac{d\Omega(W_l)}{dW_l(i,j)}(-1)$, and hence $-\frac{1}{N}\sum_{n=1}^{N} \frac{de(f^{[\mathbf{S}_{\geq l}^{i,j,n}, \mathbf{W}, x^{(n)}]}(\mathbf{W}, x^{(n)}), y^{(n)})}{dW_l(i,j)} \geq \frac{d\Omega(W_l)}{dW_l(i,j)}$. Since $\frac{d\Omega(W_l)}{dW_l(i,j)} \geq 0$, we have $|\frac{1}{N}\sum_{n=1}^{N} \frac{de(f^{[\mathbf{S}_{\geq l}^{i,j,n}, \mathbf{W}, x^{(n)}]}(\mathbf{W}, x^{(n)}), y^{(n)})}{dW_l(i,j)}| \geq |\frac{d\Omega(W_l)}{dW_l(i,j)}|$. So in particular for all $l$ and $j$ we have $||\frac{1}{N}[\sum_{n=1}^{N} \frac{de(f^{[\mathbf{S}_{\geq l}^{i,j,n}, \mathbf{W}, x^{(n)}]}(\mathbf{W}, x^{(n)}), y^{(n)})}{dW_l(i,j)}]_i||_{\frac{p}{p-1}} \geq ||[\frac{d\Omega(W_l)}{dW_l(i,j)}]_i||_{\frac{p}{p-1}} = \lambda$. So further we have:

$$\lambda \tag{63}$$

$$\leq \quad ||\frac{1}{N}[\sum_{n=1}^{N} \frac{de(f^{[\mathbf{S}_{\geq l}^{i,j,n}, \mathbf{W}, x^{(n)}]}(\mathbf{W}, x^{(n)}), y^{(n)})}{dW_l(i,j)}]_i||_{\frac{p}{p-1}} \tag{64}$$

$$= \quad \frac{1}{N}||[\sum_{n=1}^{N} g_l^{(n)[\mathbf{S}_{\geq l}^{i,j,n}, \mathbf{W}, x^{(n)}]}(j)(\sigma_l^{[\mathbf{S}_{\geq l}^{i,j,n}, \mathbf{W}, x^{(n)}]})'(z_l^{(n)}(j))x_{l-1}^{(n)}(i)]_i||_{\frac{p}{p-1}} \tag{65}$$

$$\leq \quad \frac{1}{N}||[\sum_{n=1}^{N} |g_l^{(n)[\mathbf{S}_{\geq l+1}^{i,j,n}, \mathbf{W}, x^{(n)}]}(j)||(\sigma_l^{[\mathbf{S}_{\geq l}^{i,j,n}, \mathbf{W}, x^{(n)}]})'(z_l^{(n)}(j))||x_{l-1}^{(n)}(i)]_i||_{\frac{p}{p-1}} \tag{66}$$

$$\leq \quad \frac{1}{N}B_{d\sigma,l}||[\sum_{n=1}^{N} |g_l^{(n)[\mathbf{S}_{\geq l+1}^{i,j,n}, \mathbf{W}, x^{(n)}]}(j)||x_{l-1}^{(n)}(i)]_i||_{\frac{p}{p-1}} \tag{67}$$

$$\leq \quad \frac{1}{N}B_{d\sigma,l}||[\sum_{n=1}^{N}\sum_{\mathbf{S}_{\geq l+1}} |g_l^{(n)[\mathbf{S}_{\geq l+1}, \mathbf{W}, x^{(n)}]}(j)||x_{l-1}^{(n)}(i)]_i||_{\frac{p}{p-1}} \tag{68}$$

$$\leq \quad \frac{1}{N}B_{d\sigma,l}||[\sum_{n=1}^{N}\sum_{\mathbf{S}_{\geq l+1}} |g_l^{(n)[\mathbf{S}_{\geq l+1}, \mathbf{W}, x^{(n)}]}(j)||x_{l-1}^{(n)}(i)]_i||_1 \tag{69}$$

$$= \quad \frac{1}{N}B_{d\sigma,l}\sum_{i=1}^{d_{l-1}}\sum_{n=1}^{N}\sum_{\mathbf{S}_{\geq l+1}} |g_l^{(n)[\mathbf{S}_{\geq l+1}, \mathbf{W}, x^{(n)}]}(j)||x_{l-1}^{(n)}(i)| \tag{70}$$

$$\leq \quad \frac{1}{N}B_{d\sigma,l}B_{x,l-1}\sum_{n=1}^{N}\sum_{\mathbf{S}_{\geq l+1}} |g_l^{(n)[\mathbf{S}_{\geq l+1}, \mathbf{W}, x^{(n)}]}(j)| \tag{71}$$

At line 65, we use the chain rule. Note that $x$ and $z$ do not need the $[\mathbf{S}_{\geq l}^{i,j,n}, \mathbf{W}, x^{(n)}]$ superscript because the forward evaluation of $f(\mathbf{W}, x^{(n)})$ and $f^{[\mathbf{S}_{\geq l}^{i,j,n}, \mathbf{W}, x^{(n)}]}(\mathbf{W}, x^{(n)})$ are equivalent. So, we can set $B_2 = \max_{1 \leq l \leq L} \frac{\lambda N}{B_{d\sigma,l} B_{x,l-1}}$, as required.

Finally, claim 4 is the same as in Lemma 2. The only difference in the proof is that when we invoke the box principle, we choose a specific signature $\mathbf{S}'_{\geq l+1}$ in addition to $n'$ and $j'$ such that $\sum_{i=1}^{d_l} |W_{l+1}(i, j') h_{l+1}^{(n')[\mathbf{S}'_{\geq l+1}, \mathbf{W}, x^{(n)}]}(j')| \geq \frac{d_l B_2}{d_{l+1} N 2^{D_{l+1}+..+D_L}}$. This is possible because of the induction hypothesis, which bounds the number of partial signatures $\mathbf{S}_{\geq l+1}$ by $2^{D_{l+1}+..+D_L}$. Hence we can set $D_l = (\frac{BB_{h,l+1} N D_{l+1} 2^{D_{l+1}+..+D_L}}{B_2})^p$, which completes the proof.

$\square$

*Proof of theorem 1.* Clearly, $E$ is bounded below by zero. Therefore, it has a greatest lower bound, which we call $B$. Denote $(t, t, .., t)$ by $\mathbf{d}_t$. If $\mathbf{d}$ is assumed to be fixed at $\mathbf{d}_t$, $E$ has a global minimum by lemma 1. Let $\mathbf{W}_t$ denote one such global minimum. Let $E_t$ denote the value of $E$ at $(\mathbf{d}_t, \mathbf{W}_t)$.

Now let $\mathbf{d}'$ and $\mathbf{d}''$ be two arbitrary values of $\mathbf{d}$ with $d_l' \geq d_l''$ for all $0 \leq l \leq L$. (Denote such a relation by $\mathbf{d}' \geq \mathbf{d}''$.) Then, any value that $E$ can attain with $\mathbf{d} = \mathbf{d}''$ it can attain with $\mathbf{d} = \mathbf{d}'$ because we can change any $\mathbf{d}''$-dimensional value of $\mathbf{W}$ into a $\mathbf{d}'$-dimensional value by adding $d_l' - d_l''$ units with zero fan-in and fan-out to each layer without changing $E$. In particular, this implies that $(E)_t$ is a decreasing sequence because $\mathbf{d}_{t+1} \geq \mathbf{d}_t$. Since it is also bounded below by $B$, it converges. Call its limit $C$.

Assume $C > B$. Then there exists some $(\mathbf{d}', \mathbf{W}')$ with $E(\mathbf{d}', \mathbf{W}') < C$. However, any value that $E$ can attain with $\mathbf{d} = \mathbf{d}'$ it can attain with $\mathbf{d} = \mathbf{d}_{t'}$ where $t' = \max_l d_l'$, because $\mathbf{d}_{t'} \geq \mathbf{d}'$. Therefore $C > E(\mathbf{d}', \mathbf{W}') \geq E_{t'} \geq C$. Contradiction. Therefore, $C = B$.

Now assume that for some $t$, $\mathbf{W}_t$ has a unit that has zero fan-in but not zero fan-out, or vice versa. Then by setting the non-zero fan to zero, the output of $f$ is unchanged for all $x \in \mathbb{R}^{d_0}$ and the value of $\Omega$ is reduced. Therefore, we reduce $E$, which contradicts the fact that $(\mathbf{d}_t, \mathbf{W}_t)$ is a global minimum of $E$ when $\mathbf{d}$ is fixed to $\mathbf{d}_t$. Therefore, all units in $\mathbf{W}_t$ that have zero fan-in also have zero fan-out, and vice versa.

Let $\mathbf{d}_t^{\text{proper}}$ be the proper dimensionality of $\mathbf{W}_t$ and $\mathbf{W}_t^{\text{proper}}$ be the result of removing all units with zero fan-in or fan-out from $\mathbf{W}_t$. Indeed, as we have shown, all units removed had both zero fan-in and fan-out. Assume $(\mathbf{d}_t^{\text{proper}}, \mathbf{W}_t^{\text{proper}})$ is not a local minimum of $E$. Then there exists a $\mathbf{W}'$ of dimensionality $\mathbf{d}_t^{\text{proper}}$ with $E(\mathbf{d}_t^{\text{proper}}, \mathbf{W}') < E(\mathbf{d}_t^{\text{proper}}, \mathbf{W}_t^{\text{proper}})$. When we add the zero units that were removed from $\mathbf{W}_t$ to obtain $\mathbf{W}_t^{\text{proper}}$ back into $\mathbf{W}'$, we obtain another weight parameter value we call $\mathbf{W}''$. Since $E$ is invariant under the addition and removal of units with both zero fan-in and zero fan-out, we have both $E(\mathbf{d}_t^{\text{proper}}, \mathbf{W}') = E(\mathbf{d}_t, \mathbf{W}'')$ and $E(\mathbf{d}_t^{\text{proper}}, \mathbf{W}_t^{\text{proper}}) = E(\mathbf{d}_t, \mathbf{W}_t)$. Therefore, we have $E(\mathbf{d}_t, \mathbf{W}'') < E(\mathbf{d}_t, \mathbf{W}_t)$, which contradicts that $\mathbf{W}_t$ is a global minimum of $E$ when $\mathbf{d}$ is fixed to $\mathbf{d}_t$. Therefore, $(\mathbf{d}_t^{\text{proper}}, \mathbf{W}_t^{\text{proper}})$ is a local minimum of $E$. In particular, it is a proper $E_t$-local minimum of $E$ and therefore a proper $E_0$-local minimum of $E$.

From lemma 4, we know that the set of proper $E_0$-local minima is bounded. Hence, the set $\{\mathbf{d}_t^{\text{proper}}, t \geq 0\}$ is bounded, i.e there exists some $\mathbf{d}^{\max}$ with $\mathbf{d}^{\max} \geq \mathbf{d}_t^{\text{proper}}$ for all $t$. Hence, if we denote $\max_l d_l^{\max}$ by $T$, we have $\mathbf{d}_T \geq \mathbf{d}_t^{\text{proper}}$ for all $t$ and therefore $E_T \leq E(\mathbf{d}_t^{\text{proper}}, \mathbf{W}_t^{\text{proper}})$. But $E(\mathbf{d}_t^{\text{proper}}, \mathbf{W}_t^{\text{proper}}) = E(\mathbf{d}_t, \mathbf{W}_t) = E_t$, and therefore $E_t \geq E_T$ for all $t$.

But $(E)_t$ converges to $B$ from above. Therefore $E_T = B$, therefore $E(\mathbf{d}_T, \mathbf{W}_T) = B$ and so $E$ attains its greatest lower bound which means it attains a global minimum, as required. $\square$

## 7.2 PROOF OF PROPOSITION 1

**Proposition 1.** *If all nonlinearities in a nonparametric network model except possibly $\sigma_L$ are self-similar, then the objective function 1 using a fan-in or fan-out regularizer with different regularization parameters $\lambda_1, .., \lambda_L$ for each layer is equivalent to the same objective function using the single regularization parameter $\lambda = (\prod_{l=1}^{L} \lambda_l)^{\frac{1}{L}}$ for each layer, up to rescaling of weights.*

*Proof.* Choose arbitrary positive $\lambda_1, .., \lambda_L$ and let $\lambda = (\prod_{l=1}^{L} \lambda_l)^{\frac{1}{L}}$. We have:

$$f(\mathbf{W}, x) \tag{72}$$

$$= \sigma_L.(\sigma_{L-1}.(..\sigma_2.(\sigma_1.(xW_1)W_2)..)W_L) \tag{73}$$

$$= \sigma_L.(\sigma_{L-1}.(..\sigma_2.(\sigma_1.((\prod_{l=1}^{L} \frac{\lambda_l}{\lambda})xW_1)W_2)..)W_L) \tag{74}$$

$$= \sigma_L.(\sigma_{L-1}.(..\sigma_2.((\prod_{l=2}^{L} \frac{\lambda_l}{\lambda})\sigma_1.(\frac{\lambda_1}{\lambda}xW_1)W_2)..)W_L) \tag{75}$$

$$= \sigma_L.(\frac{\lambda_L}{\lambda}\sigma_{L-1}.(..\sigma_2.(\frac{\lambda_2}{\lambda}\sigma_1.(\frac{\lambda_1}{\lambda}xW_1)W_2)..)W_L) \tag{76}$$

$$= \sigma_L.(\sigma_{L-1}.(..\sigma_2.(\sigma_1.(x(\frac{\lambda_1}{\lambda}W_1))(\frac{\lambda_2}{\lambda}W_2))..)(\frac{\lambda_L}{\lambda}W_L)) \tag{77}$$

The line-by-line explanation is as follows:

- 73  Insert the definition of $f$.

- 74  Insert a multiplicative factor of value 1.

- 75  Utilize the self-similarity of $\sigma_1$.

- 76  Repeat the previous step $L - 2$ times.

- 77  Utilize linearity.

Further, assuming we use a fan-in regularizer, we have:

$$\sum_{l=1}^{L} \lambda_l \sum_{j=1}^{d_l} ||[W_l(i,j)]_i||_p \tag{78}$$

$$= \sum_{l=1}^{L} \frac{\lambda_l}{\lambda}\lambda \sum_{j=1}^{d_l} ||[W_l(i,j)]_i||_p \tag{79}$$

$$= \sum_{l=1}^{L} \lambda \sum_{j=1}^{d_l} ||[\frac{\lambda_l}{\lambda}W_l(i,j)]_i||_p \tag{80}$$

The argument is equivalent for the fan-out regularizer.

We find that the value of the objective is preserved when we replace all regularization parameters with the same value $\lambda = (\prod_{l=1}^{L} \lambda_l)^{\frac{1}{L}}$ and rescale $W_l$ by $\frac{\lambda_l}{\lambda}$. This completes the proof.

$\square$

### 7.3 ADARAD-M

**1** **input**: $\alpha_r$: radial step size; $\alpha_\phi$: angular step size; $\lambda$: regularization hyperparameter; $\beta_{\text{arith}}$: arithmetic mixing rate; $\beta_{\text{quad}}$: quadratic mixing rate; $\epsilon$: numerical stabilizer; $\mathbf{d}^0$: initial dimensions; $\mathbf{W}^0$: initial weights; $\nu$: unit addition rate; $\nu_{\text{freq}}$: unit addition frequency; $T$: number of iterations

**2** $\phi_{\text{max}} = 0; c_{\text{max}} = 0; \mathbf{d} = \mathbf{d}^0; \mathbf{W} = \mathbf{W}^0;$

**3** **for** $l = 1$ **to** $L$ **do**

**4** set $\tilde{\phi}_l$ (angular arithmetic running average) to the zero matrix of size $d^0_{l-1} \times d^0_l$;

**5** set $\bar{\phi}_l$ (angular quadratic running average), $c_l$ (quadratic running average capacity) and $a_l$ (arithmetic running average capacity) to zero vectors of size $d^0_l$;

**6** **end**

**7** **for** $t = 1$ **to** $T$ **do**

**8** set $D^t$ to mini-batch used at iteration $t$;

**9** $\mathbf{G} = \frac{1}{|D|}\nabla_{\mathbf{W}} \sum_{(x,y)\in D^t} e(f(\mathbf{W}, x), y);$

**10** **for** $l = L$ **to** $1$ **do**

**11** $alt = \text{FALSE}$;

**12** **for** $j = d_l$ **to** $1$ **do**

**13** decompose $[G_l(i,j)]_i$ into a component parallel to $[W_l(i,j)]_i$ (call it $r$) and a component orthogonal to $[W_l(i,j)]_i$ (call it $\phi$) such that $[G_l(i,j)]_i = r + \phi$;

**14** $\bar{\phi}_l(j) = (1 - \beta_{\text{quad}})\bar{\phi}_l(j) + \beta_{\text{quad}}||\phi||_2^2; c_l(j) = (1 - \beta_{\text{quad}})c_l(j) + \beta_{\text{quad}};$

**15** $\phi_{\text{max}} = \max(\phi_{\text{max}}, \bar{\phi}_l(j)); c_{\text{max}} = \max(c_{\text{max}}, c_l(j))$ ;

**16** $[\tilde{\phi}_l(i,j)]_i = (1 - \beta_{\text{arith}})[\tilde{\phi}_l(i,j)]_i + \beta_{\text{arith}}\phi; a_l(j) = (1 - \beta_{\text{arith}})a_l(j) + \beta_{\text{arith}};$

**17** $\phi_{\text{adj}} = \dfrac{\sqrt{\frac{\phi_{\text{max}}}{c_{\text{max}}}}}{\sqrt{\frac{\bar{\phi}_l(j)}{c_l(j)}} + \epsilon} \dfrac{[\tilde{\phi}_l(i,j)]_i}{a_l(j)};$

**18** $[W_l(i,j)]_i = [W_l(i,j)]_i - \alpha_r r;$

**19** rotate $[W_l(i,j)]_i$ by angle $\alpha_\phi||\phi_{\text{adj}}||_2$ in direction $-\frac{\phi_{\text{adj}}}{||\phi_{\text{adj}}||_2}$;

**20** rotate $[\tilde{\phi}_l(i,j)]_i$ by angle $\alpha_\phi||\phi_{\text{adj}}||_2$ in direction $\frac{[W_l(i,j)]_i}{||[W_l(i,j)]_i||_2}$;

**21** shrink($[W_l(i,j)]_i, \alpha_r \lambda \frac{|D^t|}{|D|}$);

**22** **if** $l < L$ *and* $[W_l(i,j)]_i$ *is a zero vector* **then**

**23** remove column $j$ from $W_l$ and $\tilde{\phi}_l$; remove row $j$ from $W_{l+1}$ and $\tilde{\phi}_{l+1}$; remove element $j$ from $\bar{\phi}_l$, $c_l$ and $a_l$; decrement $d_l$;

**24** $alt = \text{TRUE}$;

**25** **end**

**26** **end**

**27** **if** $t = 0 \mod \nu_{\text{freq}}$ **then**

**28** $\nu' = \nu;$ `// if` $\nu \notin \mathbb{Z}$`, we can set e.g.` $\nu' = \texttt{Poisson}(\nu)$

**29** add $\nu'$ randomly initialized columns to $W_l$; add $\nu'$ zero columns to $\tilde{\phi}_l$; add $\nu'$ zero rows to $W_{l+1}$ and $\tilde{\phi}_{l+1}$; add $\nu'$ zero elements to $\bar{\phi}_l$, $c_l$ and $a_l$; $d_l = d_l + \nu';$

**30** **end**

**31** **if** $alt$ **then**

**32** **for** $j = 1$ **to** $d_{l+1}$ **do**

**33** $[\tilde{\phi}_{l+1}(i,j)]_i = [\tilde{\phi}_{l+1}(i,j)]_i - \frac{[\tilde{\phi}_{l+1}(i,j)]_i \cdot [W_{l+1}(i,j)]_i}{||[W_{l+1}(i,j)]_i||_2^2}[W_{l+1}(i,j)]_i;$

**34** **end**

**35** **end**

**36** **end**

**37** **end**

**38** **return** $\mathbf{W}$;

**Algorithm 2:** AdaRad-M with $\ell_2$ fan-in regularizer and the unit addition / removal scheme used in this paper in its most instructive (bot not fastest) order of computation.

AdaRad-M is shown in algorithm 2. The main difference in comparison to AdaRad (see algorithm 1) is that, for each fan-in, we maintain an exponential running average of the orthogonal component $[\tilde{\phi}_l(i,j)]_i$ (line 16) which we use to compute the angular shift (line 17). Hence, AdaRad-M, like Adam but unlike RMSprop and AdaRad, makes use of the principle of momentum.

One issue of note is that the running average of the orthogonal component is not itself orthogonal to the current value of the fan-in. Hence, if some multiple of it was added to the fan-in in radial-angular coordinates, it would change the length of the fan-in. This is undesirable as explained in section 3.3. Therefore, we take steps to the ensure that $[\tilde{\phi}_l(i,j)]_i$ is kept orthogonal to $[W_l(i,j)]_i$. First, whenever we rotate $[W_l(i,j)]_i$ (line 19), we rotate $[\tilde{\phi}_l(i,j)]_i$ in the same manner (line 20). Second, whenever a unit in layer $l$ and hence rows of $W_{l+1}$ and $\tilde{\phi}_{l+1}$ are deleted, we explicitly re-orthogonalize them (line 33).

## 7.4 EXPERIMENTAL DETAILS

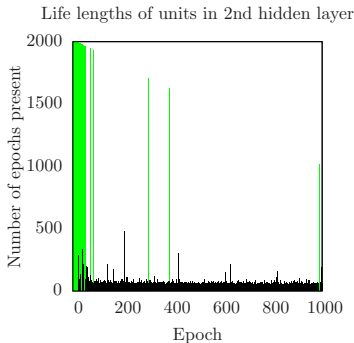

Figure 4: Length of time individual units in the second hidden layer were present during training. The x axis depicts the epoch at which a given unit was added.

In table 3, we show all hyperparameter values and related choices that were universal across all training runs and, unless specified otherwise, datasets.

### 7.4.1 PROTOCOL FOR SECTION 4.1

1. We conducted a grid search over $\lambda \in \{10^{-2}, 3*10^{-3}, 10^{-3}, 3*10^{-4}, 10^{-4}, 3*10^{-5}, 10^{-5}, 3*10^{-6}, 10^{-6}, 3*10^{-7}, 10^{-7}, 3*10^{-8}, 10^{-8}\}$ and $\alpha_\phi \in \{1, 3, 10, 30, 100, 300, 1.000, 3.000, 10.000, 30.000, 100.000\}$ for nonparametric (NP) networks using AdaRad and a single random seed, for each of the *mnist*, *rectangles-images* and *convex* datasets. By examining validation classification error (VCE) and other metrics (but not test error), we chose the single value $\alpha_\phi = 30$ for all NP experiments from now on. Further, we chose a few interesting values of $\lambda$ for each dataset. From now on, all experiments were conducted independently for each dataset.

2. We trained 10 NP networks for each chosen value of $\lambda$, with 10 different random seeds. Out of the 10 nets produced, we manually chose a single net as a typical representative by approximating the median of both network size, measured in number of weight parameters, and the test classification error (TCE) across the 10 runs. This representative, as well as the range of sizes and TCEs are shown in black in figure 2.

3. For each chosen representative, we conducted a grid search for parametric (P) networks by fixing the size of the net to the size of the representative. The grid was over $\alpha \in \{1, 3, 10, 30, 100, 300, 1.000, 3.000, 10.000, 30.000, 100.000\}$, over training algorithm (one of SGD, momentum, Nesterov momentum, RMSprop, Adam), and over whether batch normalization layers had free trainable mean and variance parameters. We introduced the last choice to more closely mimic CapNorm, which does not include free parameters. We set $\lambda = 0$ as $\ell_2$ regularization is not compatible with regular (uncapped) batch normalization. In preliminary experiments, networks trained with $\ell_2$ regularization and no batch normalization were not competitive. We used the same random seed as in step 1.

| Hyperaparameter | Value |
|---|---|
| network architecture | see figure 1 |
| number of hidden layers (not *poker*) | 2 |
| number of hidden layers (*poker*) | 4 |
| $\alpha_r$: radial step size for AdaRad (not *poker*) | $\frac{1}{50\lambda}$ |
| $\alpha_r$: radial step size for AdaRad (*poker*) | $\frac{1}{5\lambda}$ |
| $\nu$: unit addition rate for AdaRad | 1 |
| $\nu_{\text{freq}}$: unit addition frequency for AdaRad (not *poker*) | once per epoch |
| $\nu_{\text{freq}}$: unit addition frequency for AdaRad (*poker*) | ten times per epoch |
| $\beta_{\text{arith}}$: arithmetic mixing rate for AdaRad, momentum, Nesterov momentum and Adam | 0.1 |
| $\beta_{\text{quad}}$: quadratic mixing rate for AdaRad, RMSprop and Adam | 0.005 |
| $\epsilon$: numerical stabilizer for AdaRad, RMSprop and Adam | $10^{-8}$ |
| number of starting units for NP networks | 10 per hidden layer |
| $\mathbf{W}^0$: initial weights (P and NP) | $W_l^0(i,j) \sim \mathcal{N}(0, \frac{1}{\sqrt{d_{l-1}^0}})$ |
| fan-in $[W_l(i,j)]_i$ for a newly added unit $j$ | $W_l^0(i,j) \sim \mathcal{N}(0, \frac{1}{\sqrt{d_{l-1}}})$ |
| batch size | 1000 |
| batch sampling | every epoch, batches are sampled without replacement |
| type of validation (not *poker*) | one random train-valid split for each random seed |
| type of validation (*poker*) | one single random train-valid-test split for all training runs |
| train-valid split (*MNIST*) | 50.000 - 10.000 |
| train-valid split (*rectangles images*) | 10.000 - 2.000 |
| train-valid split (*convex*) | 7.000 - 1.000 |
| train-valid-test split (*poker*) | 800.000 - 125.010 - 100.000 |

Table 3: Hyperparameters and related choices.

4. We chose the 10 best performing settings from the grid search by VCE and produced 10 reruns for each setting using the same 10 random seeds as in step 2. Then we chose the best setting out of the 10 by median VCE. We depict the median as well as the range of TCE for that best setting in red in figure 2. Note that the setting that had the lowest median TCE in all cases also had the lowest median VCE.

5. We conducted a random search for P networks with 500 random settings. We chose $\alpha$ uniformly from the interval $[1, 100.000]$ in log scale. Training algorithm and type of batch normalization were chosen uniformly at random from the same sets as in step 3. The size of each hidden layer was chosen uniformly at random between the size of the corresponding layer in the largest NP representative, and 5 times that size. We used the same random seed as in step 1.

6. We chose the 10 best settings by VCE and reran them 10 times, using the same 10 random seeds as in step 2. By considering network size and median VCE, we chose 2 or 3 settings to display in blue in figure 2, including the setting with the lowest median VCE. In each case, the setting with the lowest median VCE also had the lowest median TCE.

For NP networks, we trained until the VCE had not improved for 100 epochs. Then, we rewound the last 100 epochs and kept training without adding units. After no units had been eliminated and the VCE had not improved for 100 epochs, we set $\lambda$ to zero, rewound the last 100 epochs and kept training. After the VCE had not improved for 100 epochs, we rewound again and divided the angular step size by 3. After the VCE had not improved for 5 epochs, we rewound and divided the angular step size by 3 again. We kept doing this until the angular step size was too small to change the VCE.

For P networks, we trained until the VCE had not improved for 100 epochs, then rewound and divided the step size by 3. We kept training until the VCE had not improved for 5 epochs, then rewound again and divided the step size by 3. We kept doing this until the step size was too small to change the VCE.

### 7.4.2 PROTOCOL FOR SECTION 4.3

1. We conducted a grid search over $\lambda \in \{10^{-3}, 3 * 10^{-4}, 10^{-4}, 3 * 10^{-5}, 10^{-5}, 3 * 10^{-6}, 10^{-6}, 3 * 10^{-7}, 10^{-7}\}$ and $\alpha_\phi \in \{1, 10, 100, 1.000, 10.000\}$ for NP networks using AdaRad, a single random seed and the *poker* data set. By examining VCE and other metrics (but not test error), we chose the single value $\alpha_\phi = 10$. For this value, we chose several values of $\lambda$. The size and TCE of the nets trained using those values of $\lambda$ are shown in table 2.

2. For each trained NP network shown in table 2, we trained P networks of the same size using RMSprop and each of the following step sizes: $\alpha \in \{1, 3, 10, 30, 100, 300, 1.000, 3.000, 10.000\}$. For each network size, the TCE of the network with the lowest VCE is shown in table 2. For all network sizes, the network with the lowest TCE also had the lowest VCE.

For NP networks, we trained until the VCE had not improved for 10 epochs. Then, we rewound the last 10 epochs and kept training without adding units. After no units had been eliminated and the VCE had not improved for 10 epochs, we set $\lambda$ to zero, rewound the last 10 epochs and kept training. After the VCE had not improved for 10 epochs, we rewound again and divided the angular step size by 3. After the VCE had not improved for 0.5 epochs, we rewound and divided the angular step size by 3 again. We kept doing this until the angular step size was too small to change the VCE.

For P networks, we trained until the VCE had not improved for 10 epochs, then rewound and divided the step size by 3. We kept training until the VCE had not improved for 0.5 epochs, then rewound again and divided the step size by 3. We kept doing this until the step size was too small to change the VCE.

