# Peer review of "Nonparametric Neural Networks"

_ICLR 2017 — accepted_

[Author Response · George Philipp · 01 Dec 2016 (modified: 02 Dec 2016)]
**Uploaded new revision**

Dear Reviewers,

In response to the first question posed to me, I revised the paper to include a discussion of pruning, and added several references. I would like to thank reviewer 2 for his / her feedback. I was not fully aware of how prominent this topic had become in the deep learning community recently as well as the increased use of l2 and l1 regularization.

The main changes are:

 - abstract
 - introduction
 - section 3 before 3.1
 - further background (first half)
 - conclusion

Plus, I improved the grammar / wording in a few places.

I look forward to further comments.

Best,
George

[Official Review · AnonReviewer2 · rating 5 · confidence 4 · 16 Dec 2016]
**Interesting paper with some limitations on demonstrated utility**

This paper addresses the problem of allowing networks to change the number of units that are used during training.  This is done in a simple but elegant and well-motivated way.  Units with zero input or output weights are added or removed during training, while a group sparsity norm for regularization is used to encourage unit weights to go to zero.  The main theoretical contribution is to show that with proper regularization, the loss is minimized by a network with a finite number of units.  In practice, this result does not guarantee that the resulting network will not over- or under-fit the training data, but some initial experiments show that this does not seem to be the case.

One potential advantage of approaches that learn the number of units to use in a network is to ease the burden of tuning hyperparameters.   One disadvantage of this approach (and maybe any such approach) is that it does not really solve this problem.  The network still has several hyperparameters that implicitly control the number of units that will emerge, including parameters that control how often new units are added and how rapidly weights may decay to zero.  It is not clear whether these hyperparameters will be easier or harder to tune than the ones in standard approaches.  In fairness, the authors do not claim that they have made training easier, but it is a little disappointing that this does not seem to be the case.

The authors do emphasize that they are able to train networks that use fewer units to achieve comparable performance to networks trained parametrically.  This is potentially important, because smaller networks can reduce run-time at testing, and power consumption and memory footprint, which is important on mobile devices in particular.  However, the authors do not compare experimentally to existing approaches that attempt to reduce the size of parametrically trained networks (eg., by pruning trained networks) so it is not clear whether this approach is really competitive with the best current approaches to reducing the size of trained networks.

Another potential disadvantage of the proposed approach is that the same hyperparameters control both the number of units that will appear in the network and the training time.  Therefore, training might potentially be much slower for this approach than for a parametric approach with fixed hyperparameters. In practice, many parametric approaches require methods like grid search to choose hyperparameters, which can be very slow, but in many other cases experience with similar problems can make the choice of hyperparameters relatively easy.  This means that the cost of grid search is not always paid, but the slowness of the authors’ approach may be endemic.  The authors do not discuss how this issue will scale as much larger networks are trained.  It is a concern that this approach may not be practical for large-scale networks, because training will be very slow.

In general, the experiments are helpful and encouraging, but not comprehensive or totally convincing.  I would want to see experiments on much larger problems before I was convinced that this approach can really be practical or widely useful.  

Overall, I found this to be an interesting and clearly written paper that makes a potentially useful point.  The overall vision of building networks that can grow and adapt through life-long learning is inspiring, and this type of work might be needed to realize such a vision.  But the current results remain pretty speculative.

[Official Review · AnonReviewer3 · rating 7 · confidence 4 · 21 Dec 2016 (modified: 25 Dec 2016)]
**Useful idea, limited experiments and discussion of theoretical result**

I agree with reviewer 2 on the interesting part of the paper. The idea of removing or adding units is definitely an interesting direction, that will make a model grow or shrink along the lines required by the problem and the data, not the user prior knowledge.

The authors offer an interesting theoretical result that proves that under fan out or fan in regularization the optimum of the error function is achieved for finite number of parameters - so the net does not grow indefinitely, until it over-fits perfectly the data.
That reminds me of more traditional approaches such as Lasso or Elastic Net, in which the regularization produces sparse weights. I would  have like more intuition to be given for this theorem. It is a nice result, somewhat expected (at last for me it is intuitive) and I would have liked such intuition to be given some space in the paper. For example, less discussion of prior work (that is nice too, but not as important as discussing and studying the main result of the paper) could make more room for addressing the theoretical results. Please also see below (point 2) for some suggestions.

I have a few other comments to make:

1. An interesting experiment would be to show that a model such as yours, where the nodes (neurons) are added or removed automatically can outperform a net with the same number of nodes (at the end, after complete learning), in which the size and number of nodes per layer are fixed from the start. This would prove the efficiency of the idea. This is where your method is interesting: do you save nodes that are not needed and replace them with nodes that are needed? Do you optimize performance vs. memory?

I understand that experiments along this line are given in Figure 2, with mixed results. The Figure i must say, is not very clear, but it is possible to interpret under careful inspection.

In some the non-parametric nets are doing better and others are doing worse than the parametric ones. Even in such case i could see the usefulness of the method as it helps discovering the structure. 

What i don't fully understand is why they can do better sometimes than the end net which could be trained from scratch: why is the nonparametric version of learning better than the parametric version, when the final net is known in advance? Could you give more insight?

2. Can you better discuss the meaning and implications of Theorem 1. I feel this theorem is just put there with no proper discussion. Beyond the proofs, from the Appendix, what is the key insight of the Theorem? What does it say, in plain English? To me, the conclusion seems almost natural and obvious. Is there some powerful insight? 

As i have mentioned previously, i feel this theoretical result deserves more space, with even more experiments to back it up. For example, can regularizer parameter lambda  be predicted given the data - is there a property in the data that can help guessing the right lambda? My feeling is that lambda is the key factor for determining the final net structure. Is this true? 

How much does the structure of the final net depend on the initialization? Do you get different nets if you start from different random weights? How different are they?

What happens when fan in and fan out regularizers are combined? Do you still have the same theoretical result?

I have a few additional questions:

1. Why do you say that adding zero units changes the regularizer value? For example, does L2 norm change if you add zero values?

2. Zero units are defined as having either the fan in or the fan out weights being zero. I think that what you meant is that both fan in and fan out weights are zero, otherwise you cannot remove the unit and keep the same output f. This should be clarified better I think.

I changed my rating to 7, while hoping that the authors will address my comments above.

[Official Review · AnonReviewer1 · rating 7 · confidence 3 · 27 Dec 2016]
**No Title**

This paper proposes a nonparametric neural network model, which automatically learns the size of the model during the training process. The key idea is to randomly add zero units and use sparse regularizer to automatically null out the weights that are irrelevant. The idea sounds to be a random search approach over discrete space with the help of sparse regularization to eliminate useless units. This is an important problem and the paper gives interesting results. My main comments are listed below:

What is the additional computation complexity of the algorithm? The decomposition of each fan-in weights into a parallel component and an orthogonal component and the transformation into radial-angular coordinates may require a lot of extra computation time. The authors may need to discuss the extra amount of operations relative to the parametric neural network. Furthermore, it would be useful to show some running time experiments.

It is observed that nonparametric networks return small networks on the convex dataset so that it is inferior to parametric networks. Any insight on this?

[Author Response · George Philipp · 04 Jan 2017]
**To Reviewer 2**

Dear Reviewer 2,

Thank you again for your review. Did you have a chance to look at my response? It would be great to hear your thoughts on it.

Best,
George

[Author Response · George Philipp · 13 Jan 2017]
**New version and further comments**

Dear all,

Below are some more responses to two points raised. Both responses are reflected in a new version of the paper I just uploaded.

The following has changed in the paper:

 - section 3.1 (self-similar nonlinearities) is new
 - the final paragraph of section 3 is new
 - section 7.2 (proof of new proposition) is new

The paper is now longer than 8 pages. Having looked at many other submitted papers, it appears that the 8 page requirement is not very serious. If the area chair would prefer an 8 page paper, I can move some more stuff to the appendix.

** Reducing the number of hyperparameters **

Two questions that came up throughout the reviews is whether our method reduces the number of hyperparameters and whether there is an automatic way to set lambda. 

One advantage of nonparametric networks is that instead of having one hyperparameter per layer (size) there is one hyperparameter for the entire network (lambda) that controls size. It is worth pointing out that this reduction in complexity is not arbitrary. In fact, one can prove that in a ReLU network, one regularization parameter lambda captures all the complexity of having one regularization parameter per layer because we could replace all of these regularization parameters with their geometric mean without changing the objective. (See the newly included-in-the-paper proposition 1.) Hence, nonparametric networks apportion regularization to each layer automatically.

While we don't (yet) have a great way of efficiently picking the single remaining lambda, this reduction in complexity certainly contributes to reducing hyperparameter complexity.

** Computational cost of AdaRad (in response to reviewer 1) **

(From the paper:) Using AdaRad over SGD incurs additional computational cost. However, that cost scales more gracefully than the cost of, for example, RMSprop. AdaRad normalizes each fan-in instead of each individual weight, so many of its operations scale only with the number of units and not with the number of weights in the network. In Table \ref{costTable}, we compare the costs of SGD, AdaRad and RMSprop. Further, RMSprop has a larger memory footprint than AdaRad. It requires a cache of size equal to the number of weights, whereas AdaRad only requires 2 caches of size equal to the number of neurons.

Costs (per minibatch and weight)

SGD, no $\ell_2$ shrinkage: 1 multiplication
SGD with $\ell_2$ shrinkage: 3 multiplications
AdaRad, no $\ell_2$ shrinkage: 4 multiplications
AdaRad with $\ell_2$ shrinkage: 4 multiplications
RMSprop, no $\ell_2$ shrinkage: 4 multiplications, 1 division, 1 square root
RMSprop with $\ell_2$ shrinkage: 6 multiplications, 1 division, 1 square root

Best,
George

[Author Response · George Philipp · 26 Jan 2017]
**New paper version with large dataset**

I just added a new version of the paper with experiments on a large dataset (> 1m datapoints,

[Final Decision · Program Chairs · 06 Feb 2017]
**ICLR committee final decision**

The paper presents a clean framework for optimizing for the network size during the training cycle. While the complexity of each iteration is increased, they argue that overall, the cost is significantly reduced since we do not need to train networks of varying sizes and cross-validate across them. The reviewers recommend acceptance of the paper and I am in agreement with them.